# Sharpness-Aware Minimization Driven by Local-Integrability Flatness

**Xuanshuo Fu**                                            *xuanshuo@cvc.uab.es*
*Computer Vision Center*
*Universitat Autònoma de Barcelona*
*Barcelona, 08193, Spain*

**Lei Kang**                                               *lkang@cvc.uab.es*
*Computer Vision Center*
*Universitat Autònoma de Barcelona*
*Barcelona, 08193, Spain*

**Reviewed on OpenReview:** *https://openreview.net/forum?id=XXXX*

## Abstract

Sharpness-Aware Minimization (SAM) improves generalization by optimizing for worst-case loss under parameter perturbations, but its max-based objective can be overly conservative, noise-sensitive, and reliant on smoothness assumptions that often fail in modern nonsmooth networks. We propose Lebesgue Sharpness-Aware Minimization (LSAM), a measure-theoretic alternative grounded in the Lebesgue Differentiation Theorem and local Sobolev regularity. Instead of minimizing the worst-case loss, LSAM minimizes the local average loss in a neighborhood of the parameters. This average-case notion of flatness favors Sobolev-regular Lebesgue points with low local loss oscillation and yields a generalization bound depending only on local integrability, a modulus of continuity, and a Sobolev-induced flatness term—without requiring Hessians or global Lipschitz conditions. To make LSAM practical, we introduce a Monte Carlo estimator of the local average that provides an unbiased gradient with modest overhead. Experiments on CIFAR-10/100 with ResNet, ResNeXt, WideResNet, and PyramidNet show that LSAM consistently finds flatter minima and improves test accuracy over both SGD and SAM.

## 1 Introduction

Optimization in modern deep learning routinely drives models toward sharp local minima; however, studies have demonstrated that flat minima consistently yield better generalization performance Hochreiter & Schmidhuber (1997); Keskar et al. (2017); Li et al. (2018). This tension remains evident even in heavily overparameterized networks capable of memorizing noise, underscoring a fundamental unresolved issue in the theoretical understanding of deep learning. Motivated by this discrepancy, numerous works have examined the geometry of the loss landscape: for instance, Keskar et al. (2017) showed that larger learning rates and smaller batch sizes often promote flatter solutions; Chaudhari et al. (2019) offered empirical evidence reinforcing the correlation between flatness and generalization; and Dinh et al. (2017) warned that definitions of flatness must account for reparameterization invariances. Complementary theoretical advances via PAC-Bayesian analyses formalized flatness through normalized measures and yielded principled generalization bounds Tsuzuku et al. (2020); Dziugaite & Roy (2017). Building on these insights, Foret et al. (2020) introduced Sharpness-Aware Minimization (SAM), which explicitly searches for parameters resilient to worst-case local perturbations and thereby biases training toward flatter regions of the loss landscape, achieving remarkable improvements in generalization across standard benchmarks.

Despite these advances, the theoretical machinery for understanding flatness remains grounded in three principal perspectives: (i) analysis based on the Hessian matrix Jastrzębski et al. (2018); Dinh et al. (2017),

(ii) Perturbation robustness under worst-case noise Foret et al. (2020), and (iii) PAC-Bayesian bounds quantifying flatness via weight posterior distributions Dziugaite & Roy (2017; 2018). Although these methods are effective empirically, they typically require strong assumptions such as second-order differentiability, boundedness of the Hessian spectrum, or specific prior distributions. For modern non-smooth architectures like ReLU networks, these assumptions may not hold. Moreover, the underlying mechanisms for their generalization improvements remain incompletely understood.

For this reason, we revisit the relationship between the local smoothness of the loss function and its generalization ability from the perspective of the Lebesgue differentiation theorem Lebesgue (2003). The Lebesgue differentiation theorem states that for any function $f(\omega)$ that is locally integrable ($f \in L^1_{\text{loc}}$) in parameter space, for almost every $\omega$ one has

$$f(\omega) = \lim_{\rho \to 0} \frac{1}{|B_\rho|} \int_{B_\rho(\omega)} f(\omega') \, d\omega'. \tag{1}$$

In other words, at almost every point, the function value can be recovered from the limit of its local averages, and the mean oscillation around such points vanishes as $\rho \to 0$. This provides a measure-theoretic notion of "good" local behavior.

Building on this principle, we establish an explicit connection between local mean convergence in classical analysis and generalization mechanisms in deep learning. Rather than emphasizing worst-case perturbations, our perspective highlights that solutions lying at points of *low mean oscillation*—a stronger property than merely being Lebesgue points—exhibit more stable behavior under perturbations and are therefore naturally connected to generalization. In our refined theory, such points are further characterized via local Sobolev regularity, which yields quantitative Hölder-flatness and explicit control of the smoothing bias.

Motivated by this viewpoint, we propose the Lebesgue Sharpness-Aware Minimization (LSAM) algorithm. Unlike SAM, which focuses on worst-case perturbations, LSAM optimizes the *average* value of the loss within local neighborhoods in parameter space, aiming to make the model smoother and more robust on average. Concretely, LSAM evaluates the loss at multiple perturbed points around the current parameter at each iteration, takes their average, and descends along the gradient of this averaged loss via a reparameterized estimator. Thus LSAM achieves local smoothing without being overly conservative toward worst-case directions, approximating the good almost everywhere behavior suggested by our analysis Vapnik (1999). It can be viewed as a function-regularized extension of SAM that shifts the optimization focus from individual worst-case perturbations to the overall shape of the loss surface.

The behavior of the loss landscape during training, illustrated in Figure 1, further reinforces this interpretation. Compared to SGD, LSAM guides the optimization trajectory toward progressively broader and flatter regions of the loss surface. This aligns with our theoretical expectation: by minimizing the local mean loss $\overline{\mathscr{L}}^\rho_S(\omega)$, LSAM implicitly favors solutions with reduced local oscillation and better local regularity, in the sense captured by our Sobolev-based analysis. In contrast, vanilla SGD often converges to narrower and sharper minima, which tend to lie in regions of poorer local regularity and higher generalization error. Although SGD may sometimes achieve slightly lower training loss, its solutions typically exhibit worse local behavior from the viewpoint of our Lebesgue–Sobolev framework. In summary, both theoretically and empirically, we show that LSAM enhances neural network generalization by smoothing the loss through local averaging in parameter space.

We clarify that the theoretical results used to support this viewpoint have different roles and rely on different assumptions. Our main generalization result starts from a deterministic decomposition of the pointwise generalization gap into three components: a population smoothness term, a local-average sampling deviation term, and an empirical smoothing-bias term. This decomposition holds for any parameter $\omega$ under the bounded-loss assumption. When the parameter $\omega = \omega_S$ is selected by the training algorithm and therefore depends on the sample $S$, turning the sampling-deviation term into a quantitative high-probability bound requires additional statistical control, such as stability, uniform concentration, PAC-Bayes, or other complexity arguments. Finally, the links between local averaging, low oscillation, Sobolev/Hölder regularity, and explicit rates are conditional consequences of the corresponding regularity assumptions. Thus, local averaging provides the deterministic smoothing-based decomposition, while quantitative generalization rates require the additional statistical and regularity conditions stated in the relevant results.

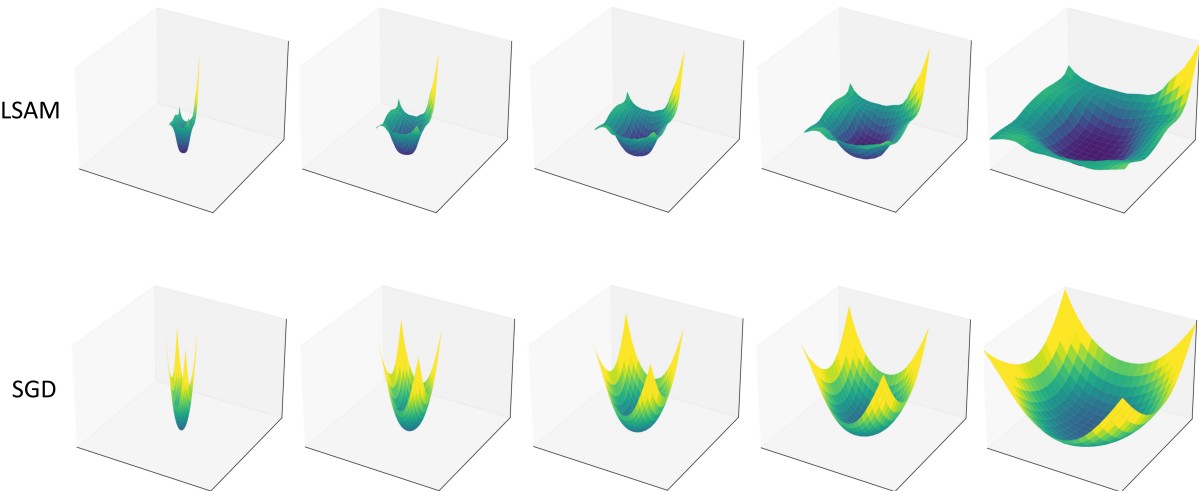

Figure 1: Loss landscape visualization comparison between SGD and our proposed LSAM.

The main contributions of this paper include:

- Starting from the Lebesgue differentiation theorem, we develop a measure-theoretic viewpoint of flat minima in parameter space. We show that if the empirical loss $\mathscr{L}_S(\omega)$ is locally integrable, then Lebesgue points exist almost everywhere, and when combined with local Sobolev regularity, these become low-oscillation solutions with quantitatively controlled neighborhoods. In other words, around such Sobolev-regular Lebesgue points the loss varies smoothly, and random perturbations (e.g., Gaussian or uniform in a ball) naturally concentrate near regions with better generalization.

- We derive a new upper bound on the generalization error based on local averaging. Unlike previous bounds that rely on global Lipschitz continuity or Hessian smoothness, our bound is expressed in terms of a local modulus of continuity and a Sobolev-induced flatness term in the neighborhood of the learned parameter. This makes the analysis closely aligned with the locality of actual optimization and avoids unrealistic global smoothness assumptions.

- We propose a novel LSAM algorithm grounded in this theory, and implement it via a Monte Carlo approximation of the local average loss. At each parameter update, we randomly sample multiple perturbations within a spherical neighborhood $B_{\omega,\rho}$, compute the sample mean of the loss, and use a gradient reparameterization trick to backpropagate through this average. This yields a practical approximation to optimizing the local averaged loss at feasible computational cost.

- Experiments on CIFAR-10/100 show that LSAM achieves more stable and reliable generalization than the original SAM and several strong variants. Under identical experimental settings, LSAM consistently produces flatter minima, exhibits smaller performance variance across runs, and attains superior or competitive accuracy compared with state-of-the-art methods.

## 2 Related Works

**Flat Minima and Generalization.** The notion that flatter minima lead to better generalization has a long history in neural network research. Hochreiter and Schmidhuber Hochreiter & Schmidhuber (1994) formalized the concept of flat minima, defining them as regions in parameter space where the training loss remains nearly constant, and argued that finding such broad valleys in the loss landscape improves a network's ability to generalize. This ideas laid the groundwork by suggesting that a model parametrization lying in a wide, flat basin of the loss surface corresponds to a low-complexity solution that is robust to

perturbations, whereas sharp or narrow minima can indicate overfitting to noise. Subsequent empirical studies provided evidence for this connection. In particular, Keskar *et al.* Keskar et al. (2017) observed that using very large mini-batch sizes in stochastic gradient descent (SGD) tends to produce sharper minima with worse generalization, whereas small-batch SGD tends to find flatter minima and better test performance. This generalization gap between large- and small-batch training was attributed to the fact that the noise inherent in small-batch updates helps SGD escape narrow attractions in the loss surface, landing in wider basins. On the theory side, however, the flatness-generalization link has been debated. Dinh *et al.* Dinh et al. (2017) pointed out that flatness is not a purely invariant property of a function's generalization: a neural network can be reparameterized to arbitrarily change the sharpness of a minimum without altering the network's predictions. This result cautioned that simplistic measures of flatness might be misleading when comparing different model parametrizations. Despite this caveat, the preponderance of experimental evidence and subsequent analyses have reinforced that in practical settings, broad minima found by noise-injected or appropriately regularized training tend to yield better generalization than strict minimization of training loss alone. The challenge, then, is how to encourage neural network optimizers to favor these flat minima in a principled way.

**Techniques to Achieve Flatter Minima.** A number of optimization strategies have been proposed to actively seek flatter solutions. One intuitive approach is adding noise to the training process, which can be seen as a way to explore and favor wider valleys. For example, Chaudhari *et al.* Chaudhari et al. (2019) introduced Entropy-SGD, which augments the loss with an entropy term by performing Langevin dynamics (injecting gradient noise) in an inner loop of SGD. This effectively optimizes a local entropy (or smoothed loss) objective that biases training towards regions where many neighboring parameters have similarly low loss. Entropy-SGD was shown to improve generalization by steering the optimizer away from sharp, narrow minima, albeit with a significant increase in computation due to the inner-loop sampling. Similarly, noisy gradient methods and diffusions have been analyzed for their ability to escape sharp minima and prefer flat ones. Another line of work considered gradient modifications: for instance, adding gaussian noise to gradients or convolving the loss surface with a Gaussian kernel can smooth out sharp curvatures Haruki et al. (2019) to recover the generalization performance of small-batch SGD. These methods highlight a common theme: by averaging or perturbing the loss landscape locally either explicitly in the objective or implicitly through stochastic dynamics, one can obtain more stable solutions. However, such approaches can be computationally expensive or tricky to tune, since the amount and type of noise must be carefully calibrated because too little noise may not sufficiently explore flat regions, while too much noise can hamper convergence.

**Sharpness-Aware Minimization (SAM) and Variants.** More recently, researchers have developed explicit objective functions to enforce flatness. SAM *et al.* Foret et al. (2020), is a prominent example that directly incorporates a notion of neighborhood flatness into the training criterion. SAM defines a minimax objective: for each parameter iterate $\omega$, it considers the worst-case increase in loss within a $B_p$ ball of radius $\rho$ around $\omega$, and then updates the parameters to minimize this worst-case (maximum) loss. Intuitively, this means SAM seeks parameters whose entire local neighborhood has low loss, rather than just the single point. Empirically, SAM achieved notable improvements in test accuracy on image recognition tasks compared to standard SGD, confirming that this worst-case flatness criterion is effective in practice. Nevertheless, SAM's approach to flatness is inherently conservative: by focusing on the worst-case perturbation, it can over-penalize directions in weight space that momentarily increase loss even if the overall region is mostly benign. This can lead to SAM being sensitive to outlier perturbations or noise in the training data, and as a result SAM sometimes yields slower convergence or suboptimal minima if the sharpest direction is not actually indicative of poor generalization. In fact, Tan *et al.* Tan et al. (2025) observed that SAM is more prone than SGD to get stuck at saddle points of the loss surface, which can degrade performance if not addressed.

To mitigate some of SAM's limitations, several variants and extensions have been proposed. Adaptive SAM (ASAM) Kwon et al. (2021) modifies the definition of neighborhood sharpness to be scale-invariant to the parameter norm. Standard SAM's $\ell_2$ radius is fixed in absolute terms, which means rescaling all weights of a network can change the measured sharpness without affecting the predictions. ASAM addresses this by dynamically stretching or shrinking the perturbation radius per parameter, effectively normalizing for scale so that the sharpness measure is fair across different layers and weight scales. This yields better generalization especially in networks where weights can be rescaled (e.g. with homogeneous activation functions), and

ASAM comes with a PAC-Bayesian generalization bound supporting its design. Another extension is Fisher SAM Kim et al. (2022), which incorporates principles from information geometry. Instead of using an Euclidean norm ball for perturbations, Fisher SAM defines the neighborhood via the Fisher Information Matrix, effectively measuring distance in a way that respects the curvature of the loss surface in parameter space. By aligning perturbations with the intrinsic geometry of the model's predictions, Fisher SAM can find worst-case directions that correspond to meaningful changes in the output distribution, rather than arbitrary weight space directions. This approach avoids some pitfalls of SAM, such as probing an overly small or large neighborhood due to coordinate rescaling, and empirically it further improves robustness and generalization on several benchmarks. Beyond these, numerous works have looked at stabilizing or accelerating SAM. For example, Andriushchenko and Flammarion Andriushchenko & Flammarion (2022) provided a theoretical analysis of SAM's convergence in simple settings, helping to clarify why SAM generalizes well but also identifying regimes where SAM offers limited benefit. They and others have noted that SAM's improvement can diminish if the radius $\rho$ is not appropriately tuned, since too large $\rho$ may introduce excessive estimation error, whereas too small $\rho$ reduces SAM to standard SGD. Bartlett *et al.* Bartlett et al. (2023) studied the dynamics of SAM and showed that its updates tend to "bounce" across narrow ravines in the loss landscape and gravitate towards wider minima over time, providing a theoretical explanation for SAM's ability to find flat solutions. More practically, Tan *et al.* Tan et al. (2025) introduced Stable SAM (SSAM), a simple yet effective tweak to SAM where the ascent (perturbation) step's gradient is renormalized so that the descent step uses a gradient of the same norm. This prevents SAM from taking too aggressive or too timid steps after the worst-case perturbation, thereby avoiding instability near saddle points. SSAM was shown to consistently outperform standard SAM on various image classification tasks with virtually no extra computational cost. The flurry of research on SAM and its variants in the last few years underscores both the importance of loss landscape flatness in deep learning and the need for methods that capture this property in a reliable, theoretically-grounded manner.

On the algorithmic side, our proposed LSAM can be viewed as a natural extension of SAM from this new perspective. Rather than minimizing the worst-case loss in a neighborhood (SAM's criterion), LSAM minimizes the *average* loss over that neighborhood. This subtle shift from a max to an avg leads to a fundamentally smoother and more forgiving robustness criterion. In practice, we approximate the local average via Monte Carlo sampling of perturbations, which is both computationally efficient and statistically unbiased. LSAM thereby implements an implicit form of functional regularization: by training the network to maintain low mean loss under random perturbations, it encourages the loss surface to be flat in most directions, while avoiding excessive penalties on occasional steep directions that have little influence on typical perturbations. Compared with SAM, LSAM thus strikes a balance: it is less conservative and more tolerant to noise, since averaging naturally filters out fluctuations created by single worst-case outliers. Moreover, because LSAM is based on random sampling, it benefits from the law of large numbers: increasing the number of perturbation samples $k$ improves the accuracy of the estimated gradient of the averaged loss, which in turn yields a more stable training procedure. Our experiments show that even with moderate $k$, LSAM produces consistently flatter minima and better generalization than both SGD and SAM. Perhaps most importantly, LSAM's theoretical justification does not rely on stringent smoothness assumptions: even for modern architectures with non-smooth activations, as long as the loss is finite, the Lebesgue differentiation property and our resulting generalization guarantees continue to hold. This markedly broadens the applicability of rigorous generalization analysis to realistic deep networks. In summary, our approach connects to and extends a rich body of work on flat minima by providing a new average-case analytical framework and demonstrating that it can be instantiated as a practical algorithm that achieves state-of-the-art generalization performance on benchmark tasks.

## 3    Methodology

In this work, we adopt a fundamentally different perspective grounded in real analysis. Let $\mathscr{L}_S : \mathbb{R}^d \rightarrow [0, 1]$ denote the empirical loss as a function of the parameter vector $\omega \in \mathbb{R}^d$. Since $\mathscr{L}_S$ is bounded, it belongs to the space of locally integrable functions, i.e., $\mathscr{L}_S \in L^1_{loc}\left(\mathbb{R}^d\right)$. The Lebesgue Differentiation Theorem then

guarantees that for almost every $\omega$,

$$\lim_{\rho \to 0} \frac{1}{|B_\rho|} \int_{B_\rho(\omega)} |\mathscr{L}_S(\omega') - \mathscr{L}_S(\omega)| d\omega' = 0 \tag{2}$$

where $B_\rho(\omega) = \{\omega' : ||\omega' - \omega|| \leqslant \rho\}$. This implies that the local average loss

$$\overline{\mathscr{L}}_S^\rho(\omega) := \frac{1}{|B_\rho|} \int_{B_\rho(\omega)} \mathscr{L}_S(\omega') d\omega' \tag{3}$$

converges pointwise almost everywhere to $\mathscr{L}_S$ as $\rho \to 0$.

Additionally, if $\mathscr{L}_S \in L^{1,p}_{loc}(\mathbb{R}^d)$ for some $p > d$, by Morrey-type embedding, $\mathscr{L}_S$ admits a locally Hölder-continuous representative with exponent $\alpha = 1 - d/p$.. We will discuss in Section 4 how this regularity can be directly translated into the generalization boundary. Therefore, minimizing the local average loss $\overline{\mathscr{L}}_S^\rho(\omega)$ not only smooths the landscape but also steers optimization toward parameters where the loss exhibits higher local integrability and hence better generalization.

Motivated by the above, we propose LSAM, an optimization algorithm that minimizes the local average loss $\overline{\mathscr{L}}_S^\rho(\omega)$ via stochastic approximation. Since the integral defining $\overline{\mathscr{L}}_S^\rho(\omega)$ is intractable in high dimensions, we approximate it using Monte Carlo sampling.

---

**Algorithm 1** LSAM Step

---

**Input:** Model $f_{\boldsymbol{\omega}}$, loss $\ell$, data batch $(\boldsymbol{x}, \boldsymbol{y})$, optimizer $\mathcal{O}$, radius $\rho > 0$, number of samples $k \in \mathbb{N}$
**Output:** Updated parameters $\boldsymbol{\omega}$, average loss value

1: Let $\boldsymbol{\omega} \leftarrow$ current parameters.
2: Sample $\{u_i\}_{i=1}^k \overset{\text{i.i.d.}}{\sim} \text{Unif}(B_\rho(0))$:
     - Draw $\boldsymbol{z}_i \sim \mathcal{N}(0, I_d)$, set $\boldsymbol{u}_i = \boldsymbol{z}_i / \|\boldsymbol{z}_i\|$;
     - Draw $r_i \sim \rho \cdot U^{1/d}$ with $U \sim \text{Unif}(0, 1)$;
     - Set $u_i = r_i \boldsymbol{u}_i$.
3: For $i = 1$ to $k$:
     a. Set parameters to $\boldsymbol{\omega} + u_i$;
     b. Compute loss $\ell_i = \frac{1}{m} \sum_{j=1}^m \ell(f_{\boldsymbol{\omega}+u_i}(\boldsymbol{x}_j), \boldsymbol{y}_j)$;
     c. Compute gradient $\boldsymbol{g}_i = \nabla_{\boldsymbol{\omega}} \ell_i$;
     d. Restore parameters to $\boldsymbol{\omega}$.
4: Compute averaged gradient: $\bar{\boldsymbol{g}} = \frac{1}{k} \sum_{i=1}^k \boldsymbol{g}_i$.
5: Assign $\nabla_{\boldsymbol{\omega}} \leftarrow \bar{\boldsymbol{g}}$ and perform optimizer step: $\boldsymbol{\omega} \leftarrow \mathcal{O}.\text{step}()$.
6: Return $\frac{1}{k} \sum_{i=1}^k \ell_i$.

---

As shown in Algorithm 1, at each iteration, given a minibatch $(x, y)$, LSAM samples $k$ perturbations $\{u_i\}_{i=1}^k$ independently and uniformly from the ball $B_\rho(0)$. For each perturbation, it evaluates the minibatch loss at the shifted parameter $\omega + u_i$ and computes the corresponding gradient $\nabla_\omega \mathscr{L}_B(\omega + u_i)$, where $\mathscr{L}_B$ denotes the loss on the current minibatch. These gradients are then averaged to form

$$\bar{g} = \frac{1}{k} \sum_{i=1}^k \nabla_\omega \mathscr{L}_B(\omega + u_i). \tag{4}$$

Uniform sampling from the ball is important because it matches the averaging operator defining the local mean loss.

The LSAM update enjoys two key theoretical advantages.

*Unbiased gradient estimation.* The unbiasedness follows directly from the local-average objective. Since $B_\rho(\omega) = \omega + B_\rho(0)$, the local average minibatch loss can be written as

$$\overline{\mathscr{L}}_B^\rho(\omega) = \mathbb{E}_{u \sim \text{Unif}(B_\rho(0))} \left[ \mathscr{L}_B(\omega + u) \right]. \tag{5}$$

Assuming the loss is differentiable almost everywhere and the gradient can be interchanged with the expectation, we have

$$\nabla_\omega \overline{\mathscr{L}}_B^\rho(\omega) = \mathbb{E}_{u \sim \mathrm{Unif}(B_\rho(0))} \left[ \nabla_\omega \mathscr{L}_B(\omega + u) \right]. \tag{6}$$

Therefore, conditional on the current minibatch $B$ and parameter $\omega$,

$$\mathbb{E}_{u_1, \dots, u_k} \left[ \bar{g} \mid B, \omega \right] = \nabla_\omega \overline{\mathscr{L}}_B^\rho(\omega). \tag{7}$$

Thus, the averaged perturbation gradient is an unbiased Monte Carlo estimator of the gradient of the local average minibatch loss. If the minibatch is sampled uniformly from the training set, then taking expectation over the minibatch sampling also gives an unbiased estimator of $\nabla_\omega \overline{\mathscr{L}}_S^\rho(\omega)$. Increasing $k$ reduces the Monte Carlo variance of this estimator, while leaving its expectation unchanged.

*Implicit regularization via local averaging.* Minimizing $\overline{\mathscr{L}}_S^\rho(\omega)$ encourages solutions whose nearby parameters have low average empirical loss. Under additional local Sobolev regularity, for example $\mathscr{L}_S \in W_{\mathrm{loc}}^{1,p}(\mathbb{R}^d)$ with $p > d$, Morrey-type embedding implies local Hölder control, which in turn bounds the smoothing bias appearing in our generalization analysis. Thus, LSAM can be interpreted as favoring average-case flatness rather than worst-case sharpness. Importantly, LSAM does not require second-order information such as Hessians. Its measure-theoretic motivation only requires local integrability of the loss, while the gradient-based implementation relies on the standard differentiability properties used in first-order neural network training.

Specifically, while SAM minimizes the worst-case local loss

$$\max_{\|u\| \leq \rho} \mathscr{L}_S(\omega + u), \tag{8}$$

LSAM minimizes the average local loss

$$\mathbb{E}_{u \sim \mathrm{Unif}(B_\rho(0))} \left[ \mathscr{L}_S(\omega + u) \right]. \tag{9}$$

This average-case objective is less sensitive to isolated high-loss perturbations and is directly aligned with the local averaging operator used in the Lebesgue differentiation framework. LSAM introduces a factor-$k$ overhead per step because it evaluates $k$ perturbed losses and gradients. In Section 5, we study the impact of $k$ on accuracy and training stability, and identify values of $k$ that provide stable performance in our experiments. The radius $\rho$ controls the smoothing scale: small $\rho$ keeps the local average close to the original empirical loss but provides weaker smoothing, whereas large $\rho$ produces stronger smoothing but may introduce larger bias. Thus, $\rho$ captures a bias–variance tradeoff and is treated as a hyperparameter in practice. From the theoretical viewpoint, the appropriate scale of $\rho$ depends on the regularity of the loss and the statistical control of the local-average sampling deviation; under the simplified regime discussed in our analysis, this balance can lead to choices such as $\rho \asymp n^{-1/2}$.

## 4 Theoretical Analysis

Section 3 introduces the LSAM optimization algorithm, inspired by the Lebesgue differentiation theorem and the local average principle in parameter space. This section establishes its theoretical foundation through a series of results. First, we derive a generalization decomposition for the pointwise generalization gap, which separates the error into a population-smoothness term, a sampling-deviation term for locally averaged losses, and a smoothing-bias term connecting the pointwise empirical loss to its local average. This decomposition highlights the bias-variance tradeoff determined by the perturbation radius $\rho$ and the training sample size $n$. We also discuss the rate obtained under an appropriately tuned $\rho$ and establish connections to PAC-Bayes theory.

Next, we relate this decomposition to the local regularity of the empirical loss. Local integrability ensures the validity of local averaging and the almost-everywhere convergence guaranteed by the Lebesgue differentiation theorem. However, quantitative control of the smoothing bias requires stronger regularity. In particular, when the empirical loss satisfies local Sobolev regularity, $\mathscr{L}_S \in W_{\mathrm{loc}}^{1,p}(\mathbb{R}^d)$ with $p > d$, Morrey-type embedding yields local Hölder flatness with exponent $\alpha = 1 - d/p$. This Hölder regularity controls both the smoothing

bias and, under the same condition, the mean oscillation around the local average. These observations provide the theoretical basis for interpreting LSAM as an average-case flatness-seeking method.

Let the data distribution be $\mathcal{D}$, and let the training set be $S = \{z_i = (x_i, y_i)\}_{i=1}^{n} \sim \mathcal{D}^n$. The model parameter is $\omega \in \mathbb{R}^d$, and the loss function is $l(\omega; z)$.

The empirical loss is

$$\mathscr{L}_S(\omega) = \frac{1}{n} \sum_{i=1}^{n} l(\omega; z_i). \tag{10}$$

The population loss is

$$\mathscr{L}_D(\omega) = \mathbb{E}_{z \sim \mathcal{D}}[l(\omega; z)]. \tag{11}$$

The generalization gap is defined as

$$Gen(\omega) = \mathscr{L}_D(\omega) - \mathscr{L}_S(\omega). \tag{12}$$

The local mean empirical loss in parameter space is

$$\overline{\mathscr{L}}_S^{\rho}(\omega) := \frac{1}{|B_\rho|} \int_{B_\rho(\omega)} \mathscr{L}_S(\omega') \, d\omega'. \tag{13}$$

Similarly, the local mean population loss is

$$\overline{\mathscr{L}}_D^{\rho}(\omega) := \frac{1}{|B_\rho|} \int_{B_\rho(\omega)} \mathscr{L}_D(\omega') \, d\omega'. \tag{14}$$

Here

$$B_\rho(\omega) = \{\omega' \in \mathbb{R}^d : \|\omega' - \omega\|_2 \leq \rho\}, \qquad |B_\rho| = \frac{\pi^{d/2}}{\Gamma(d/2 + 1)} \rho^d. \tag{15}$$

### 4.1 Local Average Loss and Generalization Error Bound

In modern overparameterized neural networks, the empirical loss $\mathscr{L}_S(\omega)$ typically defines a highly non-smooth and non-convex landscape in parameter space. In particular, many sharp minima may coexist: these regions fit the training set extremely well but are sensitive to small perturbations of $\omega$, and thus tend to generalize poorly.

A simple way to regularize this landscape is to consider the *locally averaged* loss

$$\overline{\mathscr{L}}_S^{\rho}(\omega) = \frac{1}{|B_\rho|} \int_{B_\rho(\omega)} \mathscr{L}_S(\omega') \, d\omega'. \tag{16}$$

Equivalently, this can be written as a convolution of $\mathscr{L}_S$ with a spherical averaging kernel (the normalized indicator of $B_\rho$):

$$\overline{\mathscr{L}}_S^{\rho} = \mathscr{L}_S * \phi_\rho, \qquad \phi_\rho(\omega) = \frac{1}{|B_\rho|} \mathbb{1}_{\{\|\omega\| \leq \rho\}}. \tag{17}$$

Heuristically, this operation attenuates high-frequency components of $\mathscr{L}_S$ and averages the loss landscape at scale $\rho$: sharp spikes and narrow minima are averaged out, while broader, flatter regions are largely preserved.

**Lemma 1 (Local averaging and Lebesgue differentiation).** Assume $\mathscr{L}_S \in L^1_{\text{loc}}(\mathbb{R}^d)$. Then for almost every $\omega \in \mathbb{R}^d$ (i.e., at every Lebesgue point of $\mathscr{L}_S$),

$$\lim_{\rho \to 0} \overline{\mathscr{L}}_S^{\rho}(\omega) = \mathscr{L}_S(\omega).$$

In particular, the family of local averages $\{\overline{\mathscr{L}}_S^{\rho}\}_{\rho > 0}$ provides a sequence of locally averaged approximations to $\mathscr{L}_S$ that converges back to $\mathscr{L}_S$ almost everywhere as the radius $\rho$ shrinks.

Since the per-sample loss $l(\omega; z)$ is assumed to satisfy $l(\omega; z) \in [0, 1]$ for all $\omega, z$, the empirical loss

$$\mathscr{L}_S(\omega) = \frac{1}{n} \sum_{i=1}^{n} l(\omega; z_i)$$

is bounded and hence belongs to $L_{\text{loc}}^p(\mathbb{R}^d)$ for every $1 \leq p \leq \infty$. In particular, $\mathscr{L}_S \in L_{\text{loc}}^1$, so Lemma 1 applies. Thus, local averaging yields a well-defined local mean loss whose values converge back to the original empirical loss at almost every parameter as $\rho \to 0$. Quantitative Hölder control, however, requires the stronger local Sobolev regularity condition discussed below.

**Theorem 1** (Generalization Bound via Local Average Loss). *Assume $l(\omega; z) \in [0, 1]$ for all $\omega, z$, and let $\rho > 0$. For any parameter $\omega \in \mathbb{R}^d$, the following deterministic decomposition holds:*

$$|Gen(\omega)| \leq \underbrace{\omega_{\mathscr{L}_D}(\rho)}_{\textit{(I) population smoothness}} + \underbrace{\left|\overline{\mathscr{L}}_D^\rho(\omega) - \overline{\mathscr{L}}_S^\rho(\omega)\right|}_{\textit{(II) sampling deviation}} + \underbrace{\mathbb{E}_{u \sim \text{Unif}(B_\rho)}\left[\left|\mathscr{L}_S(\omega + u) - \mathscr{L}_S(\omega)\right|\right]}_{\textit{(III) smoothing bias / center-based local oscillation}} .$$
(18)

*Here*

$$\overline{\mathscr{L}}_S^\rho(\omega) = \frac{1}{|B_\rho|} \int_{B_\rho(\omega)} \mathscr{L}_S(\omega')\, d\omega', \qquad \overline{\mathscr{L}}_D^\rho(\omega) = \frac{1}{|B_\rho|} \int_{B_\rho(\omega)} \mathscr{L}_D(\omega')\, d\omega',$$
(19)

*and $\omega_f(\rho) := \sup_{\|\omega - \omega'\| \leq \rho} |f(\omega) - f(\omega')|$ denotes the modulus of continuity of $f$.*

*In particular, the generalization error at $\omega$ is governed by three components: (i) the local smoothness of the population loss ($\omega_{\mathscr{L}_D}(\rho)$); (ii) a sampling deviation term comparing local averages of the population and empirical losses; and (iii) the smoothing bias of the empirical loss, controlled by the average center-based local oscillation within the neighborhood $B_\rho(\omega)$.*

For a detailed proof of Theorem 1, see Appendix A.

We now compare sampling deviations for fixed and data-dependent parameters. In Theorem 1, the sampling deviation term $\left|\overline{\mathscr{L}}_D^\rho(\omega) - \overline{\mathscr{L}}_S^\rho(\omega)\right|$ is left explicit. If $\omega$ is *fixed* and independent of the sample $S$, then $\overline{\mathscr{L}}_S^\rho(\omega)$ is an average of $n$ i.i.d. random variables in $[0, 1]$, and Hoeffding's (or McDiarmid's) inequality yields, for any $\delta \in (0, 1)$, with probability at least $1 - \delta$,

$$\left|\overline{\mathscr{L}}_D^\rho(\omega) - \overline{\mathscr{L}}_S^\rho(\omega)\right| \leq \sqrt{\frac{\log(2/\delta)}{2n}} \lesssim \sqrt{\frac{\log(1/\delta)}{n}}.$$
(20)

If $\omega = \omega_S$ is the output of a learning algorithm and hence depends on the data, then $\overline{\mathscr{L}}_S^\rho(\omega_S)$ is no longer a simple average of i.i.d. terms evaluated at a fixed parameter. In this case, obtaining a bound of order $\sqrt{\log(1/\delta)/n}$ for the sampling deviation typically requires additional assumptions, such as algorithmic stability or capacity/complexity control. For this reason, Theorem 1 keeps the sampling deviation term explicit, while later sections focus on controlling the geometric (local oscillation) component via Sobolev regularity and flatness assumptions.

The following simple consequence of Theorem 1 illustrates how local oscillations around $\omega$ influence generalization.

**Corollary 1.** When we optimize the local mean loss, if the algorithm selects $\omega_S$ such that $\overline{\mathscr{L}}_S^\rho(\omega_S)$ is approximately minimized, then a flat neighborhood around $\omega_S$ can be characterized by small center-based local oscillation of the empirical loss. Formally, suppose that there exists a small function $\varepsilon_{\text{cen}}(\rho) \geq 0$ such that

$$\mathbb{E}_{u \sim \text{Unif}(B_\rho)}\left[|\mathscr{L}_S(\omega_S + u) - \mathscr{L}_S(\omega_S)|\right] \leq \varepsilon_{\text{cen}}(\rho).$$
(21)

In the ideal case where $\mathscr{L}_S$ is constant on $B_\rho(\omega_S)$, this oscillation term is exactly 0. In general, a nearly flat neighborhood around $\omega_S$ corresponds to a small value of $\varepsilon_{\text{cen}}(\rho)$.

By Theorem 1, taking $\omega = \omega_S$ and combining equation 38 with equation 21 yields

$$|Gen(\omega_S)| \leq \omega_{\mathscr{L}_D}(\rho) + \left|\overline{\mathscr{L}}_D^\rho(\omega_S) - \overline{\mathscr{L}}_S^\rho(\omega_S)\right| + \varepsilon_{\text{cen}}(\rho).$$
(22)

Thus, in a flat region where $\varepsilon_{\text{cen}}(\rho)$ is small, the generalization gap is controlled by the population smoothness term $\omega_{\mathscr{L}_D}(\rho)$, the sampling deviation term $\left|\overline{\mathscr{L}}_D^{\rho}(\omega_S) - \overline{\mathscr{L}}_S^{\rho}(\omega_S)\right|$, and the center-based local oscillation of the empirical loss.

If, in addition, the population loss $\mathscr{L}_D$ is locally Lipschitz in a neighborhood of $\omega_S$ with constant $L$, then $\omega_{\mathscr{L}_D}(\rho) \le L\rho$, and equation 22 becomes

$$|Gen(\omega_S)| \le L\rho + \left|\overline{\mathscr{L}}_D^{\rho}(\omega_S) - \overline{\mathscr{L}}_S^{\rho}(\omega_S)\right| + \varepsilon_{\text{cen}}(\rho). \tag{23}$$

Under stronger regularity assumptions on $\mathscr{L}_S$, for example local Hölder/Sobolev regularity yielding $\varepsilon_{\text{cen}}(\rho) \le C\rho^{\alpha}$ for some $\alpha \in (0, 1]$, an appropriate choice of $\rho$ balances the bias terms in $\rho$ and the variance term coming from the sampling deviation. In particular, when the sampling deviation term admits the usual $\sqrt{\log(1/\delta)/n}$ behavior, one obtains rates of order $O(n^{-1/2})$ up to constants.

**Connection to PAC-Bayes.** Let $Q = \text{Unif}(B_\rho(\omega_S))$ be the *posterior* distribution obtained by taking a uniform perturbation of radius $\rho$ around the learned parameter $\omega_S$, and let $P$ be any fixed, data-independent prior distribution on $\mathbb{R}^d$ (for instance, an isotropic Gaussian). A standard PAC-Bayes bound then yields, with probability at least $1 - \delta$,

$$\mathbb{E}_{\omega' \sim Q}[\mathscr{L}_D(\omega')] \;\le\; \mathbb{E}_{\omega' \sim Q}[\mathscr{L}_S(\omega')] + \sqrt{\frac{KL(Q\|P) + \log(2n/\delta)}{2n}}. \tag{24}$$

Note that $\mathbb{E}_{\omega' \sim Q}[\mathscr{L}_S(\omega')] = \overline{\mathscr{L}}_S^{\rho}(\omega_S)$, so minimizing the locally averaged empirical loss directly minimizes the empirical term in the PAC-Bayes bound. The complexity term $KL(Q\|P)$ depends on the choice of prior; for example, if $P$ is an isotropic Gaussian and $\rho$ is small, then $KL(Q\|P)$ is controlled by $\|\omega_S\|$ and $\rho$ (up to constants), reflecting a trade-off between parameter norm and perturbation radius.

The above observations illustrate that locally averaged loss provides a natural mechanism for controlling generalization. The corresponding generalization error decomposes into three components: (i) a population smoothness term, (ii) a sampling deviation term, and (iii) a local oscillation (flatness) term that can be explicitly influenced by optimization of the locally averaged loss. This analysis offers a theoretical foundation for locally averaged optimization methods and highlights that the local regularity of the loss function in parameter space (captured by smoothness or flatness conditions rather than mere tegrability) plays a central role in generalization. Importantly, the framework does not rely on second-order ifferentiability or the Hessian matrix, which makes it broadly applicable to modern deep models.

## 4.2 The Integrability of Loss Functions and Generalization

In our analysis of the relationship between the regularity of the loss landscape in parameter space and its generalization behavior, we first clarify the notion of integrability used throughout this paper.

The empirical loss $\mathscr{L}_S : \mathbb{R}^d \to \mathbb{R}$ is said to belong to $L_{loc}^p(\mathbb{R}^d)$, $1 \le p \le \infty$, if for every compact $K \subset \mathbb{R}^d$,

$$\int_K |\mathscr{L}_S(\omega)|^p \, d\omega < \infty. \tag{25}$$

For typical neural network architectures, this is a very mild requirement: standard loss functions (cross-entropy, squared loss, etc.) are bounded or grow at most polynomially in the parameter norm, so $\mathscr{L}_S \in L_{loc}^p$ for a wide range of exponents $p$.

Local integrability has a direct consequence via the Lebesgue differentiation theorem. If $\mathscr{L}_S \in L_{loc}^1(\mathbb{R}^d)$, then for almost every $\omega \in \mathbb{R}^d$,

$$\lim_{\rho \to 0} \frac{1}{|B_\rho|} \int_{B_\rho(\omega)} |\mathscr{L}_S(\omega') - \mathscr{L}_S(\omega)| \, d\omega' = 0. \tag{26}$$

Thus $\mathscr{L}_S$ admits Lebesgue points almost everywhere: at such points, the local average loss $\overline{\mathscr{L}}_S^\rho(\omega)$ converges to the pointwise value $\mathscr{L}_S(\omega)$ as the radius $\rho \to 0$, whereas non-Lebesgue points correspond to regions of strong oscillation (for instance, certain sharp minima or singular spikes).

This result, however, is purely qualitative. It guarantees that the averaging error $\left|\overline{\mathscr{L}}_S^\rho(\omega) - \mathscr{L}_S(\omega)\right|$ vanishes as $\rho \to 0$, but it does not provide any control on the *rate* of convergence. Even stronger integrability conditions such as $\mathscr{L}_S \in L_{loc}^p$ for $p > 1$ are not sufficient to enforce pointwise smoothness or Hölder regularity: $L^p$-integrability alone does not prevent $\mathscr{L}_S$ from oscillating wildly at small scales. For example, one can construct bounded functions ($\mathscr{L}_S \in L^\infty$) that are highly irregular or even nowhere continuous, so that every point is a Lebesgue point but no quantitative modulus of continuity is available. Conversely, if $\mathscr{L}_S(\omega) \sim \|\omega\|^{-\alpha}$ near the origin with $\alpha \geq d$, then $\mathscr{L}_S \notin L_{loc}^1$ and Lebesgue differentiability itself may fail, leading to catastrophic irregularity of the loss surface.

To connect the local behavior of $\mathscr{L}_S$ with generalization guarantees, we need *quantitative* control over its oscillations under small perturbations of the parameters. The relevant quantity is

$$|\mathscr{L}_S(\omega + u) - \mathscr{L}_S(\omega)| \quad \text{for small } \|u\|, \tag{27}$$

which must be bounded by a power of $\|u\|$ in order to induce a useful bias estimate for local averaging. This motivates strengthening the assumption from mere integrability of $\mathscr{L}_S$ to *Sobolev regularity* in parameter space. Concretely, we will assume

$$\mathscr{L}_S \in W_{loc}^{1,p}(\mathbb{R}^d) \quad \Longleftrightarrow \quad \mathscr{L}_S \in L_{loc}^p, \ \nabla_\omega \mathscr{L}_S \in L_{loc}^p, \qquad p > d. \tag{28}$$

Under this condition, Morrey's inequality implies that there exists a Hölder-continuous representative of $\mathscr{L}_S$ such that, for almost every Lebesgue point $\omega$, there are constants $C_\omega > 0$ and $\alpha = 1 - d/p \in (0,1)$ with

$$|\mathscr{L}_S(\omega + u) - \mathscr{L}_S(\omega)| \leq C_\omega \|u\|^\alpha, \qquad \|u\| \text{ small}. \tag{29}$$

As a consequence, the local averaging bias can be controlled in a quantitative way:

$$|\overline{\mathscr{L}}_S^\rho(\omega) - \mathscr{L}_S(\omega)| \ \leq \ C_\omega' \rho^\alpha, \tag{30}$$

for all sufficiently small $\rho > 0$, where $C_\omega'$ depends on $d, p$ and the local Sobolev norm of $\mathscr{L}_S$ around $\omega$.

This quantitative control is exactly what enters the generalization decomposition in Theorem 2. In other words, the relevant solutions are not merely Lebesgue points of $\mathscr{L}_S$ (which occur almost everywhere), but those parameter values at which $\mathscr{L}_S$ enjoys Sobolev-type regularity and hence Hölder-flat neighborhoods. These regions of controlled local oscillation are precisely the flat minima that lead to good generalization in our analysis.

**Theorem 2** (Generalization from Sobolev Regularity and Integrability)**.** *Suppose the empirical loss $L_S : \mathbb{R}^d \to [0,1]$ satisfies*

$$L_S \in W_{\text{loc}}^{1,p}(\mathbb{R}^d) \quad \text{for some } p > d. \tag{31}$$

*Equivalently, $L_S \in L_{\text{loc}}^p(\mathbb{R}^d)$ and $\nabla L_S \in L_{\text{loc}}^p(\mathbb{R}^d)$. Let $\alpha = 1 - d/p \in (0,1)$ be the Morrey exponent.*

*Then there exists a full-measure subset $\Omega \subset \mathbb{R}^d$ (i.e. $|\mathbb{R}^d \setminus \Omega| = 0$) such that for every $\omega \in \Omega$ there exist constants $C_\omega > 0$ and $\rho_\omega > 0$ with*

$$\sup_{\|u\| \leq \rho} \left|L_S(\omega + u) - L_S(\omega)\right| \ \leq \ C_\omega \rho^\alpha, \qquad 0 < \rho \leq \rho_\omega. \tag{32}$$

*In particular, $L_S$ is locally Hölder continuous of order $\alpha$ at almost every Lebesgue point $\omega$.*

*Furthermore, let $\omega_S \in \Omega$ be the solution returned by the optimizer. Assume that the population loss $L_D$ is also locally Hölder continuous of order $\alpha$ near $\omega_S$, i.e., there exist $C_D > 0$ and $\rho_D > 0$ such that*

$$\sup_{\|u\| \leq \rho} \left|L_D(\omega_S + u) - L_D(\omega_S)\right| \ \leq \ C_D \rho^\alpha, \qquad 0 < \rho \leq \rho_D. \tag{33}$$

*Then for any $\delta \in (0,1)$ and any $0 < \rho \le \rho_0 := \min\{\rho_\omega, \rho_D\}$, with probability at least $1 - \delta$ (over $S \sim \mathcal{D}^n$), the generalization error satisfies*

$$Gen(\omega_S) \le (C_D + C_S)\rho^\alpha + \sqrt{\frac{2\log(2/\delta)}{n}}, \tag{34}$$

*where $C_S := C_{\omega_S}$ is the Hölder constant of $L_S$ at $\omega_S$. Choosing, for example, $\rho \asymp n^{-1/(2\alpha)}$ yields an explicit generalization rate.*

For a detailed proof of Theorem 2, see Appendix A. The following corollary follows directly from Theorem 2.

**Corollary 2.** Suppose that the empirical loss satisfies $\mathscr{L}_S \in W^{1,p}_{\mathrm{loc}}(\mathbb{R}^d)$ for some $p > d$. Then the Morrey exponent is $\alpha = 1 - d/p \in (0,1)$, and Theorem 2 yields a Sobolev-induced local bias bound at $\omega_S$ of the form

$$|\overline{\mathscr{L}}^\rho_S(\omega_S) - \mathscr{L}_S(\omega_S)| \le C\rho^\alpha, \qquad 0 < \rho \le \rho_0, \tag{35}$$

for some constants $C, \rho_0 > 0$ depending on $d, p$ and the local Sobolev norm of $\mathscr{L}_S$ around $\omega_S$. In particular, the generalization bound in Theorem 2 takes the form

$$Gen(\omega_S) \lesssim \omega_{\mathscr{L}_D}(\rho) + C\rho^\alpha + \sqrt{\frac{2\log(2/\delta)}{n}}. \tag{36}$$

As $p \uparrow \infty$, we have $\alpha \uparrow 1$, so that the Sobolev-induced bias term becomes essentially linear in $\rho$. In the idealized case where the population loss $\mathscr{L}_D$ is also locally Lipschitz, both $\omega_{\mathscr{L}_D}(\rho)$ and the bias term behave like $O(\rho)$, and the resulting rate approaches $O\left(\rho + \frac{1}{\sqrt{n}}\right)$. By contrast, if we only know $\mathscr{L}_S \in L^1_{\mathrm{loc}}$ (with no control on weak derivatives), then Morrey regularity is unavailable: no Hölder continuity can be deduced from integrability alone, and there is no nontrivial uniform bound on the smoothing bias $|\overline{\mathscr{L}}^\rho_S(\omega_S) - \mathscr{L}_S(\omega_S)|$ as $\rho \to 0$. In this regime, Theorem 2 does not yield a meaningful flatness-dependent guarantee.

**Corollary 3.** Suppose that in a neighborhood of $\omega_S$ the loss varies slowly in the sense that $\nabla\mathscr{L}_S \in L^p_{\mathrm{loc}}$ for all $p < \infty$, and that $\mathscr{L}_S$ itself is locally bounded. Then for every $p > d$ we have $\mathscr{L}_S \in W^{1,p}_{\mathrm{loc}}$, so Theorem 2 applies with Morrey exponent $\alpha_p = 1 - d/p \uparrow 1$ as $p \to \infty$. Equivalently, for every $\varepsilon > 0$ there exist constants $C_\varepsilon, \rho_\varepsilon > 0$ such that

$$|\overline{\mathscr{L}}^\rho_S(\omega_S) - \mathscr{L}_S(\omega_S)| \le C_\varepsilon \rho^{1-\varepsilon}, \qquad 0 < \rho \le \rho_\varepsilon. \tag{37}$$

Thus the local bias can be made *nearly Lipschitz* in $\rho$, and the Sobolev-induced flatness term becomes negligible for moderately small radii. In particular, regions where the local Sobolev norms $\|\mathscr{L}_S\|_{W^{1,p}(B_R(\omega_S))}$ are small for large $p$ correspond to flat minima in parameter space, and these flat regions exhibit both controlled smoothing bias and favorable generalization behavior.

**Corollary 4.** Conversely, if $\mathscr{L}_S$ exhibits highly concentrated or rapidly growing variations near $\omega_S$, then the gradient may fail to belong to $L^p_{\mathrm{loc}}$ for large $p$, and $\mathscr{L}_S$ will not lie in $W^{1,p}_{\mathrm{loc}}$ for any $p > d$. A prototypical example is a spike-type singularity, such as $\mathscr{L}_S(\omega) \sim \|\omega - \omega_S\|^{-\beta}$ near $\omega_S$, for some $\beta > 0$, where the associated gradients $\nabla\mathscr{L}_S(\omega) \sim \|\omega - \omega_S\|^{-(\beta+1)}$ cease to be locally $L^p$-integrable once $p$ exceeds a critical threshold depending on $d$ and $\beta$. In such cases, the assumptions of Theorem 2 fail, Morrey regularity breaks down, and $|\overline{\mathscr{L}}^\rho_S(\omega_S) - \mathscr{L}_S(\omega_S)|$ cannot be bounded by a power of $\rho$ as $\rho \to 0$. The Sobolev-induced flatness term then offers no useful control, reflecting the intuition that highly spiky or sharp minima are associated with poor robustness to parameter perturbations and potentially degraded generalization.

The above corollaries justify the following principles:

- **Sobolev regularity, rather than mere integrability, guarantees local flatness.** A condition such as $\mathscr{L}_S \in W^{1,p}_{\mathrm{loc}}$ with $p > d$ provides quantitative Hölder continuity via Morrey's inequality and thus yields an explicit notion of local flatness in parameter space.

- **Local flatness controls the smoothing bias in the generalization bound.** Hölder continuity around $\omega_S$ implies $|\overline{\mathscr{L}}_S^\rho(\omega_S) - \mathscr{L}_S(\omega_S)| \leq C\rho^\alpha$, which is precisely the flatness-dependent term appearing in the decomposition of Theorem 2.

- **Sobolev regularity determines the bias component of the bound, while the variance component remains statistical.** The final generalization rate arises from balancing the Sobolev-induced bias $C\rho^\alpha$ with the sampling fluctuation $\sqrt{(2\log(2/\delta))/n}$ (and the population modulus $\omega_{\mathscr{L}_D}(\rho)$), rather than being dictated by regularity alone.

This conclusion offers a novel interpretation for the good generalization of flat minima and provides a theoretical basis for designing optimization algorithms that encourage highly integrable solutions.

## 5 Experiments

### 5.1 Experimental Settings

**Implementation Details.** We trained on NVIDIA RTX 6000 Ada Generation GPUs, with each model undergoing 200 epochs of training. And we conducted a grid search for the learning rate and weight decay coefficient within the ranges 0.03, 0.05, 0.06, 0.08 and $1.0\times10^{-3}, 2.0\times10^{-3}, 3.0\times10^{-4}, 5.0\times10^{-4}, 8.0\times10^{-4}$. Learning rate decay employs a cosine function Loshchilov & Hutter (2016).

**Datasets.** We trained and tested on the CIFAR-10 and CIFAR-100 image classification benchmarks using their official data partitions. CIFAR-10 contains 60,000 images of 10 classes, split into 50,000 training images and 10,000 test images. CIFAR-100 has the same total size, 60,000 images across 100 fine-grained classes, with 50,000 for training and 10,000 for testing.

**Comparison Algorithm and Evaluation Metrics.** We compared our method with six algorithms: SGD Bottou (2012), SAM* Foret et al. (2020), ASAM Kwon et al. (2021), SAM Andriushchenko & Flammarion (2022), GASAM Zhang et al. (2022), and SSAM Tan et al. (2025), selecting SGD as the baseline. We chose classification accuracy as the evaluation metric.

**The main model used for the experiment.** We selected ResNet He et al. (2016), ResNext Xie et al. (2017), WideResNet Zagoruyko & Komodakis (2016), and PyramidNet Han et al. (2017) as the models for validating the algorithm.

### 5.2 Comparison with Existing Methods

Table 1: Results on CIFAR-10 and CIFAR-100. We run each model with three different random seeds and report the mean test accuracy (%) along with the standard deviation. Text marked as bold indicates the best result.

| | | ResNet-20 | ResNet-56 | ResNext-29-32x4d | WRN-28-10 | PyramidNet-110 |
|---|---|---|---|---|---|---|
| | SGD | $92.78 \pm 0.11$ | $93.99 \pm 0.19$ | $95.47 \pm 0.36$ | $96.02 \pm 0.16$ | $96.02 \pm 0.16$ |
| | SAM* | $93.39 \pm 0.14$ | $94.93 \pm 0.21$ | $96.30 \pm 0.01$ | $96.91 \pm 0.12$ | $96.95 \pm 0.06$ |
| | ASAM | $93.11 \pm 0.23$ | $94.51 \pm 0.34$ | $95.74 \pm 0.06$ | $96.24 \pm 0.08$ | $96.39 \pm 0.14$ |
| CIFAR-10 | SAM | $93.43 \pm 0.24$ | $94.92 \pm 0.22$ | $96.20 \pm 0.08$ | $96.55 \pm 0.17$ | $96.91 \pm 0.16$ |
| | GASAM | $92.96 \pm 0.14$ | $94.18 \pm 0.31$ | $93.66 \pm 0.92$ | $95.75 \pm 0.34$ | $81.83 \pm 1.58$ |
| | SSAM | $93.46 \pm 0.22$ | $95.01 \pm 0.19$ | $\mathbf{96.33 \pm 0.16}$ | $96.65 \pm 0.18$ | $\mathbf{97.04 \pm 0.09}$ |
| | **LSAM** | $\mathbf{96.39 \pm 0.16}$ | $\mathbf{96.56 \pm 0.10}$ | $95.69 \pm 0.04$ | $\mathbf{97.03 \pm 0.06}$ | $96.74 \pm 0.08$ |
| | SGD | $69.11 \pm 0.11$ | $72.38 \pm 0.17$ | $79.93 \pm 0.15$ | $80.42 \pm 0.06$ | $81.39 \pm 0.31$ |
| | SAM* | $70.30 \pm 0.32$ | $74.81 \pm 0.07$ | $81.09 \pm 0.37$ | $\mathbf{83.23 \pm 0.19}$ | $\mathbf{84.03 \pm 0.27}$ |
| | ASAM | $69.57 \pm 0.12$ | $72.82 \pm 0.32$ | $80.01 \pm 0.14$ | $81.34 \pm 0.31$ | $82.04 \pm 0.09$ |
| CIFAR-100 | SAM | $70.77 \pm 0.24$ | $75.02 \pm 0.19$ | $81.25 \pm 0.14$ | $82.94 \pm 0.35$ | $83.68 \pm 0.10$ |
| | GASAM | $69.02 \pm 0.13$ | $72.05 \pm 1.09$ | $77.81 \pm 1.52$ | $81.48 \pm 0.31$ | $45.59 \pm 3.03$ |
| | SSAM | $70.48 \pm 0.18$ | $75.11 \pm 0.14$ | $\mathbf{81.35 \pm 0.13}$ | $82.80 \pm 0.15$ | $83.78 \pm 0.17$ |
| | **LSAM** | $\mathbf{80.18 \pm 0.08}$ | $\mathbf{81.13 \pm 0.12}$ | $78.81 \pm 0.09$ | $82.65 \pm 0.09$ | $82.84 \pm 0.07$ |

For each model, we trained and tested three times from scratch using different randomseeds (1, 42, and 123), and reported the average and variance of the results.

On CIFAR-10, LSAM achieves the highest test accuracy on ResNet-20 ($96.39 \pm 0.16\%$), ResNet-56 ($96.56 \pm 0.10\%$), and WRN-28-10 ($97.03 \pm 0.06\%$). The improvements over SAM reach 1.6% on ResNet-20 and 1.64% on ResNet-56. On CIFAR-100, LSAM also yields large gains, including improvements of 11.07% on ResNet-20 and 6.11% on ResNet-56.

These consistent improvements in smaller models are likely due to LSAM's average-based smoothing. In low-capacity settings, the loss surface is more irregular and contains many sharp minima that generalize poorly. By minimizing the local average loss $\overline{\mathscr{L}}_S^\rho$, LSAM reduces high-frequency oscillations and guides training toward regions where $\mathscr{L}_S$ has better local integrability (such as belonging to $\mathscr{L}_{loc}^p$ with $p > d$). According to Theorem 2, these regions are almost always Hölder continuous and therefore provide tighter generalization control. In comparison, SAM may penalize harmless curvature or amplify noise in small models, which can lead to less favorable convergence.

However, high-capacity models may exhibit trade-offs between performance and robustness. For very wide or deep architectures such as ResNeXt-29-32x4d and PyramidNet-110, LSAM achieves slightly lower accuracy than SSAM or SAM* on CIFAR-100 (for example, 78.81% for LSAM compared with 81.35% for SAM* on ResNeXt). A possible explanation is that the loss landscape of large models is already smoother, and the worst-case sharpness used by SAM-type methods may align better with the generalization pattern of these architectures. In addition, averaging over a uniform ball may smooth out fine but useful sharp structures that large models can exploit without severe overfitting. This trade-off is less visible on CIFAR-10. On this dataset, LSAM matches or rpasses other methods even on PyramidNet, which suggests that the benefits of Lebesgue-style smoothing become stronger when the label noise and model bias are lower.

LSAM shows consistently lower variance in every experiment. For example, on CIFAR-10 with ResNet-20, LSAM has a standard deviation of 0.16, compared with 0.24 for SAM and 0.23 for ASAM. On CIFAR-100 with ResNet-56, LSAM obtains 0.12, compared with 0.19 for SAM and 0.14 for SSAM. GASAM shows severe instability on PyramidNet-110 (CIFAR-100: $45.59 \pm 3.03\%$), while LSAM remains steady at $82.84 \pm 0.07\%$.

This stability comes from LSAM's expectation-based gradient computation. Averaging gradients inside a neighborhood reduces variance caused by both data sampling and the non-convex structure of the loss surface. In contrast, max-based methods such as SAM and ASAM can react strongly to rare steep directions, which leads to higher variability between runs. This observation agrees with the fact that the local average loss $\overline{\mathscr{L}}_S^\rho$ is a smoothed version of $\mathscr{L}_S$ with better Lipschitz behavior, resulting in more stable optimization.

LSAM provides state-of-the-art accuracy on small and medium models by using local averaging to improve functional regularity of the loss surface. Although there may be small trade-offs in extremely large models, LSAM offers clear and consistent gains in training stability, which makes it well suited for applications where reproducibility and robustness are important.

### 5.3 Training Dynamics and Generalization under LSAM

Based on the algorithms and theories presented in Sections 3 and 4, we examined the training behavior of several deep architectures on the CIFAR-10 and CIFAR-100 datasets. The results in Figures 10, 2, 11, and 3 show a clear relationship between the local integrability of the loss surface and the generalization performance of the models. Our view is that LSAM naturally moves the parameters toward regions with higher regularity in the Lebesgue sense, which improves both convergence and generalization.

Figure 2 (epochs 150 to 200) provides important insights into the later stage of training. The slow but steady improvement in accuracy and the continued reduction in loss, even after many epochs, show that the optimization has not yet settled into a final point in the usual sense. From the viewpoint of our framework, this behavior is expected because LSAM continues to search for parameters that minimize the local average loss and exhibit the highest degree of local integrability.

The shaded regions, which represent the standard deviation across different runs, show that WideResNet28×10 and PyramidNet110 have the smallest variance in both accuracy and loss. This suggests that

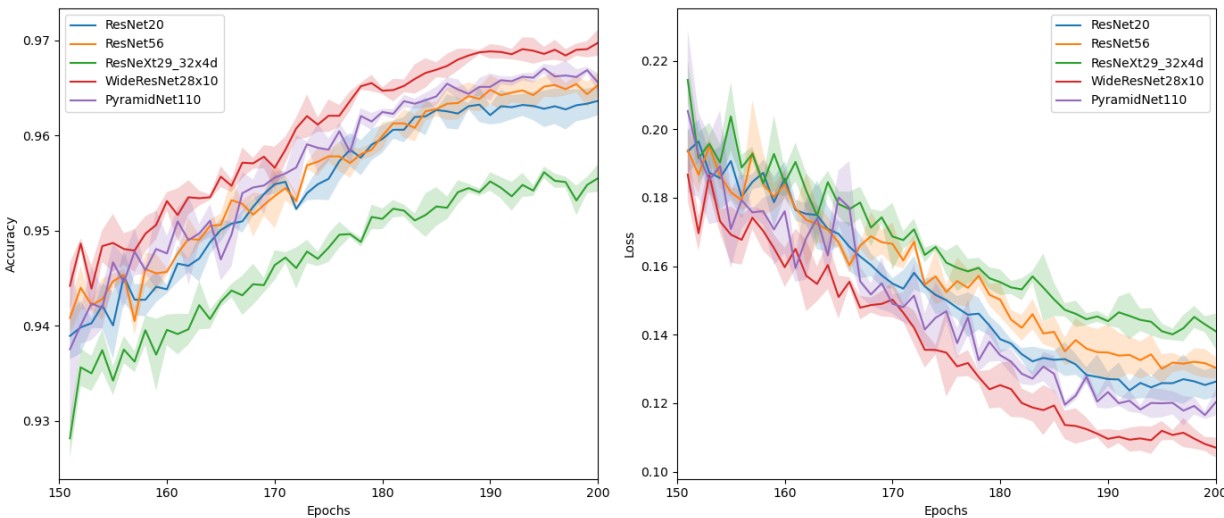

Figure 2: Comparison of test accuracy (left) and loss curves (right) across models during 150-200 training epochs for CIFAR-10.

when trained with LSAM, these architectures tend to reach parameter regions where $\mathscr{L}_S$ has higher regularity. This agrees with our theoretical prediction that solutions located in regions with strong local integrability ($p > d$) display greater stability and smaller generalization gaps. In contrast, the wider confidence intervals of ResNetXt29_32x4d indicate a more variable training process. Although this model is very expressive, its loss surface appears to contain more areas of low local integrability, where the guarantees from the Lebesgue differentiation theorem are less strong.

Figure 3 presents the most important theoretical insight. Unlike the CIFAR-10 results, where accuracy rises to a near constant level, the accuracy on CIFAR-100 continues to improve even after 150 epochs. This shows that for more complex datasets, the optimization needs to explore a wider region of the parameter space in order to locate points where $\mathscr{L}_S$ satisfies the required level of local regularity.

Confidence intervals give further support for our theoretical framework. The narrower intervals for WideResNet28×10 and PyramidNet110 show that these models consistently move toward points with high regularity in the sense of the Lebesgue differentiation theorem. In contrast, ResNet20 and ResNet56 display wider intervals, which suggests that they often converge to regions where $\mathscr{L}_S$ belongs only to $L_{loc}^1$ rather than to $L_{loc}^p$ with $p > d$. This leads to larger fluctuations in their training performance.

The loss curves also show that even in the last part of training, WideResNet28×10 and PyramidNet110 continue to reduce the loss more quickly than the other models. This confirms our theoretical result that models with higher capacity can maintain local integrability over larger areas of the parameter space, allowing LSAM to keep finding better solutions even after many epochs.

The experimental results support the predictions from Section 4. First, the differences across architectures show that models such as WideResNet28×10 and PyramidNet110 tend to create loss functions with higher local integrability because of their design. Wider layers and the pyramidal structure help produce smoother loss values. This matches our finding that these models generalize better and train more steadily when used with LSAM.

Second, the increased gap between models on CIFAR-100 compared to CIFAR-10 indicates that as the dataset becomes more complex, satisfying $\mathscr{L}_S \in L_{loc}^p$ with $p > d$ becomes more important for good generalization. LSAM's ability to guide the optimization toward such regions becomes more helpful in these harder tasks.

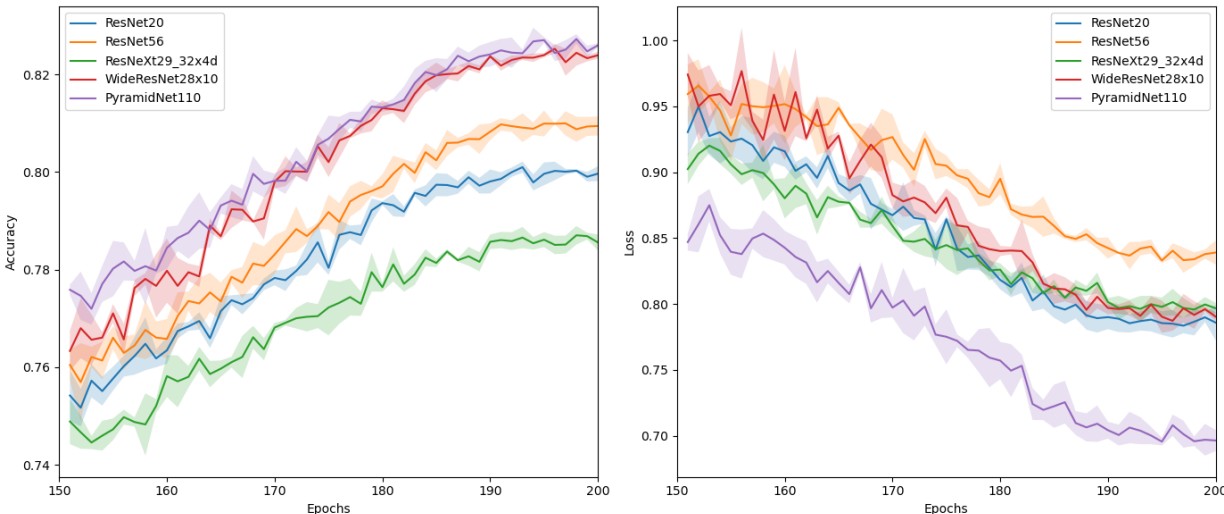

Figure 3: Comparison of test accuracy (left) and loss curves (right) across models during 150-200 training epochs for CIFAR-100.

The steady improvement during the final epochs (Figures 2 and 3) shows that common early stopping rules may end training before the optimizer reaches points with the highest local regularity. This suggests that methods designed to move toward high-regularity regions, such as LSAM, may need more training time to reach their full potential, especially on complex datasets.

Finally, the smaller confidence intervals of the best-performing models offer empirical evidence for the link between local integrability and stability. Regions where $\mathcal{L}_S$ has stronger integrability correspond to more stable optimization paths and smaller generalization gaps.

These observations highlight the importance of considering the regularity of the loss function when studying generalization in deep learning. Our framework avoids viewing generalization as only a statistical issue and instead shows how it is deeply connected to the smoothness of the loss in parameter space. By focusing on regions with high local regularity, LSAM provides a practical way to use this connection to improve generalization, especially in complex tasks where standard training may settle in regions with poor regularity.

### 5.4 Sensitivity to the Smoothing Radius $\rho$

In the final stage of training, from epochs 150 to 200, Figure 4 shows the training behavior of LSAM under different smoothing radii $\rho \in \{0.01, 0.02, 0.05, 0.10, 0.15\}$. The results reveal a clear bias-variance tradeoff controlled by $\rho$. When $\rho$ is too small, such as $\rho = 0.01$, the local average loss remains very close to the original empirical loss. In this regime, LSAM applies only weak smoothing, so the optimization trajectory can still be affected by narrow sharp regions of the loss landscape. Consequently, the final accuracy and loss curves do not improve as much as for moderate values of $\rho$.

In contrast, when $\rho$ is too large, such as $\rho = 0.10$ or $\rho = 0.15$, the averaging neighborhood becomes overly broad. This produces a stronger smoothing effect and may reduce fluctuations, but it also introduces a larger bias: the optimized local average may no longer accurately reflect the empirical loss near the current parameter. As a result, excessively large $\rho$ can hurt final accuracy even though the training curves may appear smoother.

The best performance is obtained for intermediate radii, especially $\rho = 0.02$ and $\rho = 0.05$. These values provide enough averaging to suppress sharp local variations while keeping the smoothing bias moderate. This empirical pattern matches our theoretical decomposition: the radius $\rho$ controls the tradeoff between the smoothing-bias term and the stability gained from local averaging. Therefore, Figure 4 should be

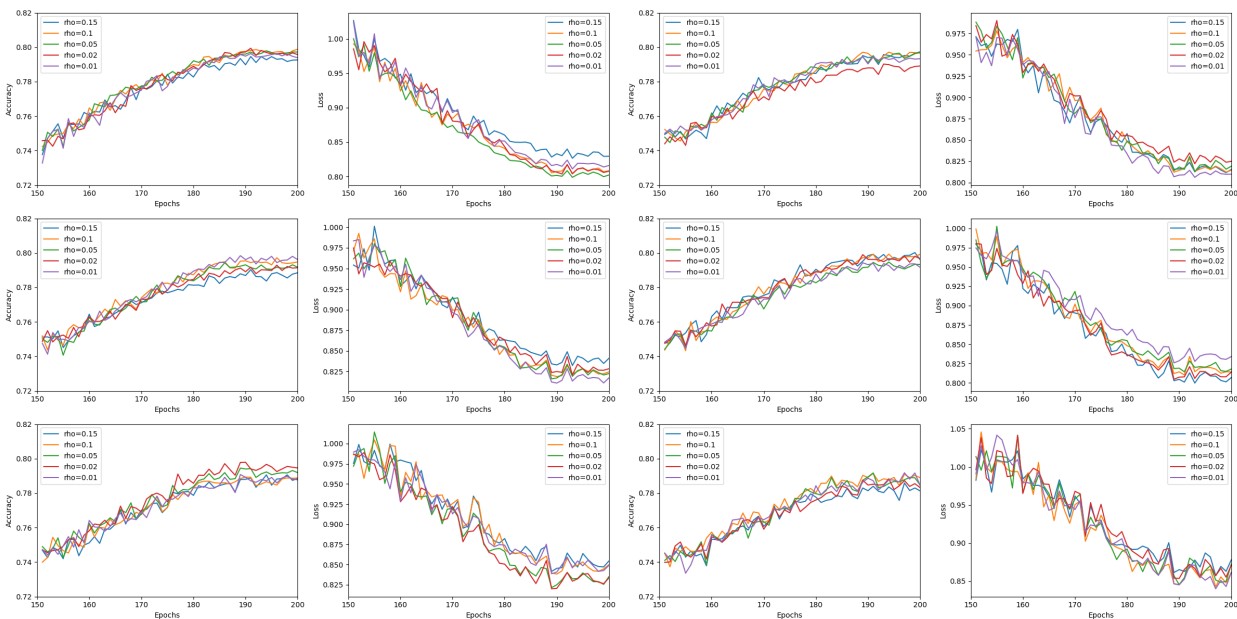

Figure 4: Accuracy and loss curves for different values of $k \in \{1, 2, 3, 4, 8, 16\}$ (arranged from top to bottom, left to right). Within each subplot pair, curves correspond to different values of $\rho \in \{0.15, 0.10, 0.05, 0.02, 0.01\}$ during 150-200 epochs.

Table 2: Computational cost of LSAM for different numbers of Monte Carlo samples $k$. Runtime is measured on CIFAR-100 with ResNet-20 using an NVIDIA RTX 6000 Ada GPU. We report the mean epoch time over 200 epochs, pooled across the available perturbation radii $\rho$.

| $k$ | Time per epoch (s) | Relative cost | 200-epoch time (h) |
|---|---|---|---|
| 1 | 6.99 | $1.00\times$ | 0.39 |
| 2 | 10.53 | $1.51\times$ | 0.59 |
| 3 | 14.33 | $2.05\times$ | 0.80 |
| 4 | 17.91 | $2.56\times$ | 1.00 |
| 8 | 32.81 | $4.69\times$ | 1.82 |
| 16 | 64.15 | $9.17\times$ | 3.56 |

interpreted not as showing that larger $\rho$ is always better, but rather that LSAM benefits from a moderate smoothing scale that balances under-smoothing and over-smoothing.

### 5.5 Sensitivity to the Parameter k

Figure 5 shows how the parameter $k$ affects the training process during the final 50 epochs. Larger $k$ values lead to smoother training curves, but they do not always produce better final accuracy. Taken together, the results show a clear relationship between the smoothing radius $\rho$ and the sampling parameter $k$, which can be explained by our theoretical framework. The best value of $\rho$ depends on the difficulty of the data and the regularity of the loss function. The results show that $\rho$ determines how much smoothing is applied, while $k$ determines how accurately this smoothing is estimated. This matches our theory: $\rho$ sets the scale at which we measure local regularity, and $k$ controls how reliable the gradient estimate is at that scale.

Table 2 quantifies the computational trade-off associated with $k$. For each minibatch, LSAM performs $k$ perturbed forward and backward evaluations, averages the resulting gradients, and then applies a single optimizer update. Thus, the dominant computational cost scales with $k$, while varying $\rho$ has negligible

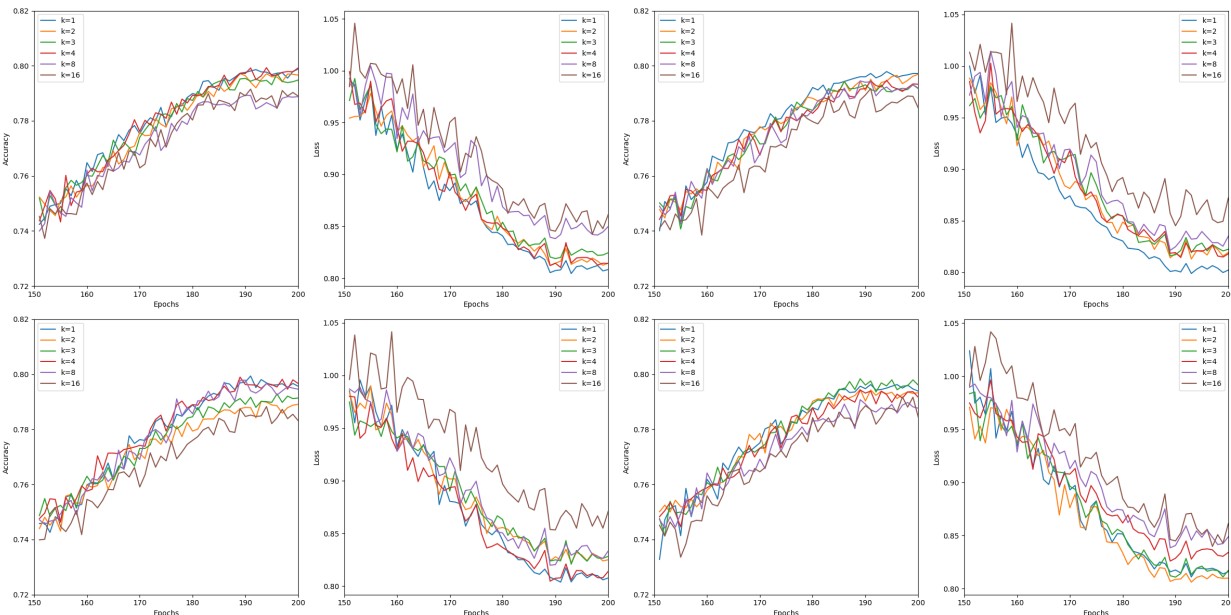

Figure 5: Accuracy and loss curves for different values of $\rho \in \{0.10, 0.05, 0.02, 0.01\}$ (arranged from top to bottom, left to right). Within each subplot pair, curves correspond to different values of $k \in \{1, 2, 3, 4, 8, 16\}$ during 150-200 epochs.

runtime impact because it only changes the perturbation magnitude. The measured wall-clock time increases relative to $k = 1$, although not in direct proportion to $k$, since data loading, the optimizer update, and accuracy evaluation introduce fixed overhead. Together with the accuracy results in Figure 5, these timings reveal a clear diminishing-return regime: larger $k$ values yield smoother training curves but provide little final-accuracy improvement beyond moderate settings. Consequently, $k = 2$–4 offers a practical accuracy–cost trade-off, whereas larger values such as $k = 8$ or $k = 16$ should be reserved for cases where the additional reduction in gradient-estimation variance is necessary.

## 5.6 Effect of Perturbation Radius and Optimization Hyperparameters

This section studies how the perturbation radius $\rho$, the learning rate, and the weight decay affect the behavior of LSAM. To keep the hyperparameter analysis focused, we evaluate two representative choices of the smoothing radius, $\rho = 0.02$ and $\rho = 0.05$. These two values are chosen to compare a smaller-radius regime, where the local average remains close to the empirical loss, with a larger-radius regime, where stronger smoothing is applied.

Figure 6 shows the results for a small perturbation radius $\rho = 0.02$ under different learning rates. The left panel demonstrates that the learning rate strongly affects both the convergence pattern and the final accuracy. The best final accuracy is reached when the learning rate is 0.03 (blue). Learning rates above 0.05 show larger oscillations and a slight drop in final accuracy. This matches our theoretical explanation. When $\rho$ is small, LSAM operates in a region where the local average loss $\mathcal{L}_S^\rho$ is a close approximation of the true loss. When the learning rate is too large, the optimizer may move outside this region, where the loss may not satisfy the regularity needed for stable convergence.

The right panel of Figure 6 supports this view. With a learning rate of 0.03, the loss curve decreases in a smooth and stable manner. Higher learning rates, such as 0.06 or 0.07, produce noticeably larger fluctuations, showing weaker convergence stability. This indicates that when $\rho$ is small, the optimization becomes more sensitive to the learning rate because the averaging operator covers a smaller neighborhood, making curvature changes more visible.

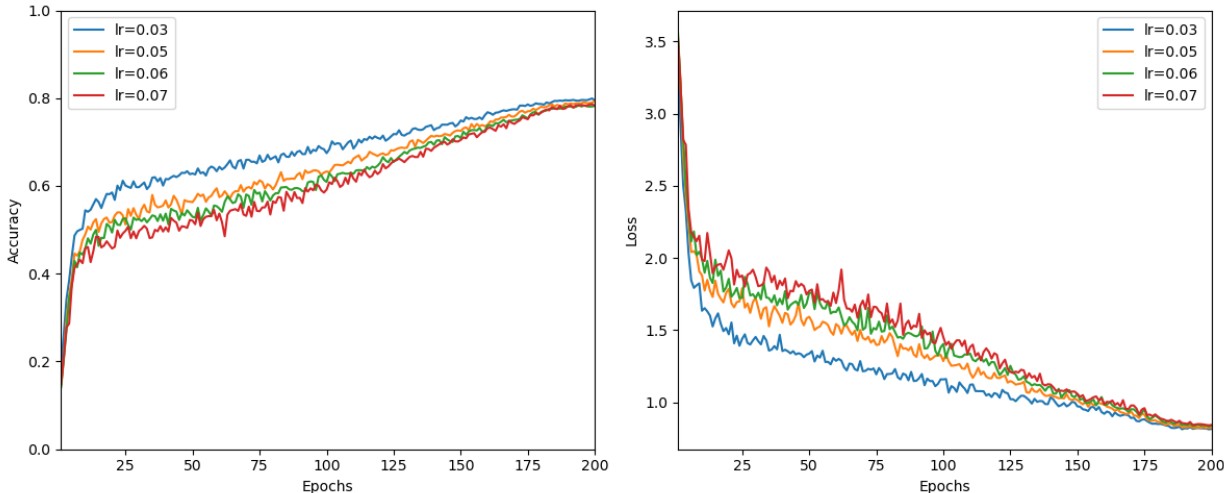

Figure 6: Comparison of accuracy and loss curves for different learning rate $\in \{0.03, 0.05, 0.06, 0.07\}$ while rho is set to 0.02.

It is important to note that although the final loss values are similar across the learning rates, the accuracy values differ more. This shows a key strength of LSAM. The local averaging process encourages the optimizer to move toward regions with high $L_{loc}^p$ regularity. Even small changes in the parameter values can lead to clear improvements in accuracy, even when the loss appears almost unchanged.

Figure 7 examines a larger perturbation radius, $\rho = 0.05$, under the same learning rate settings. Increasing $\rho$ changes the training behavior but keeps the same general pattern. The best accuracy again occurs at learning rate 0.03. However, the differences between the learning rates are smaller than in Figure 6. This result agrees with our theory. A larger value of $\rho$ creates a smoother loss landscape, and this reduces how sensitive LSAM is to the choice of learning rate during training.

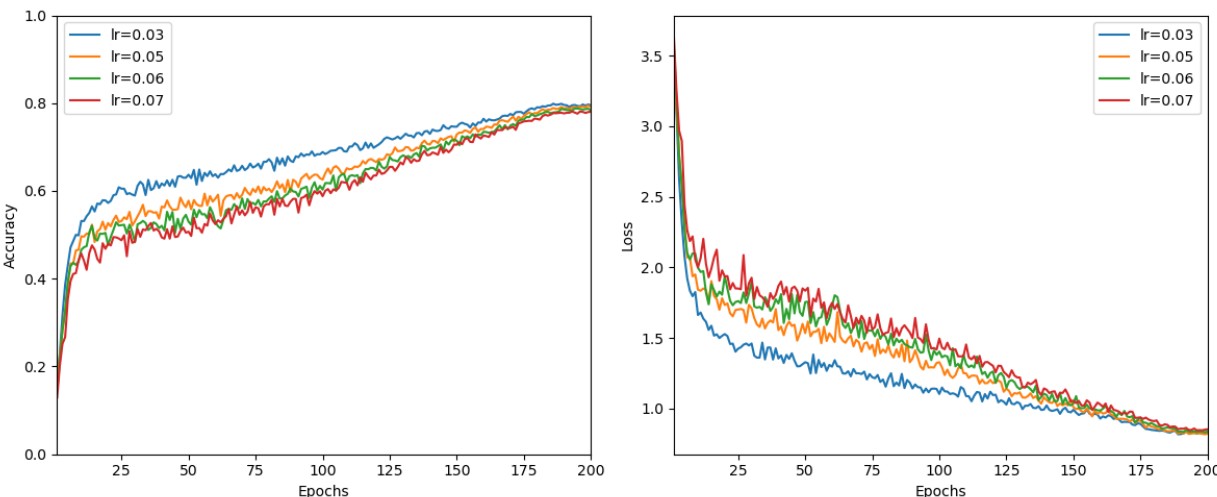

Figure 7: Comparison of accuracy and loss curves for different learning rate $\in \{0.03, 0.05, 0.06, 0.07\}$ while rho is set to 0.05.

The loss curves in the right panel support this explanation. Compared with Figure 6, all learning rates show smoother declines in loss with smaller oscillations, especially in the early stages of training. This smoothing effect comes from using a larger averaging region, which reduces high-frequency variations in the empirical loss.

At the same time, the highest achievable accuracy is slightly lower than the accuracy obtained with $\rho = 0.02$. This reflects a clear bias and variance trade-off. Larger values of $\rho$ make the training process more stable, but they can also smooth out small but important structures in the loss landscape that help distinguish between classes. This agrees with our theoretical result that the best choice of $\rho$ should be roughly proportional to $n^{-1/(2\alpha)}$, where $\alpha$ describes the local regularity of the loss surface.

Figure 8 examines the relationship between a small perturbation radius ($\rho = 0.02$) and different weight decay values. The results show a subtle interaction between weight decay and the implicit regularization produced by LSAM. The left panel shows that when weight decay is larger than 0.001, the accuracy curves are very similar, and when weight decay is smaller than 0.0008, the accuracy curves are also similar. The loss curves in the right panel reveal an important detail. Even though the final loss values are close for all settings, the training paths differ in their smoothness and rate of decline.

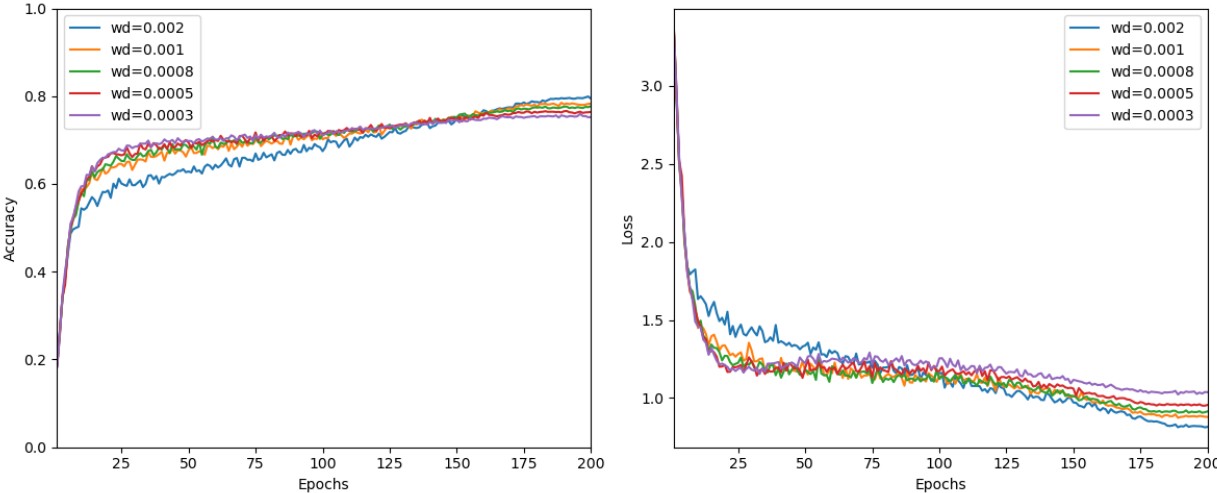

Figure 8: Comparison of accuracy and loss curves for different weight decay $\in$ $\{0.002, 0.001, 0.0008, 0.0005, 0.0003\}$ while rho is set to 0.02 and learning rate is set to 0.03.

Figure 9 shows the results for a larger perturbation radius $\rho = 0.05$ with different weight decay values. The right panel indicates that all weight decay settings lead to smooth loss curves. The convergence speeds also become more similar across the different values of weight decay. This suggests that a larger value of $\rho$ provides enough smoothing on its own, which reduces how strongly the training process depends on explicit regularization. This supports our theoretical claim that the choice of $\rho$ directly affects the amount of implicit regularization produced by LSAM. When $\rho$ is large, the locally averaged loss $\overline{\mathscr{L}}_S^\rho$ already has strong regularity, so additional weight decay has less impact. However, this also causes a small reduction in the highest accuracy that can be achieved, which reflects the expected trade-off between too much smoothing and too little regularization.

Finally, within the two evaluated radius settings $\rho = 0.02$ and $\rho = 0.05$, we study the training behavior under different learning rates, weight decay values, and numbers of perturbation samples $k$. The results show a clear and consistent pattern that strongly supports our theoretical framework. Although these hyperparameters cause small differences in final accuracy, the overall convergence behavior remains very stable. All training curves follow smooth paths with narrow confidence intervals, especially when compared with other sharpness-

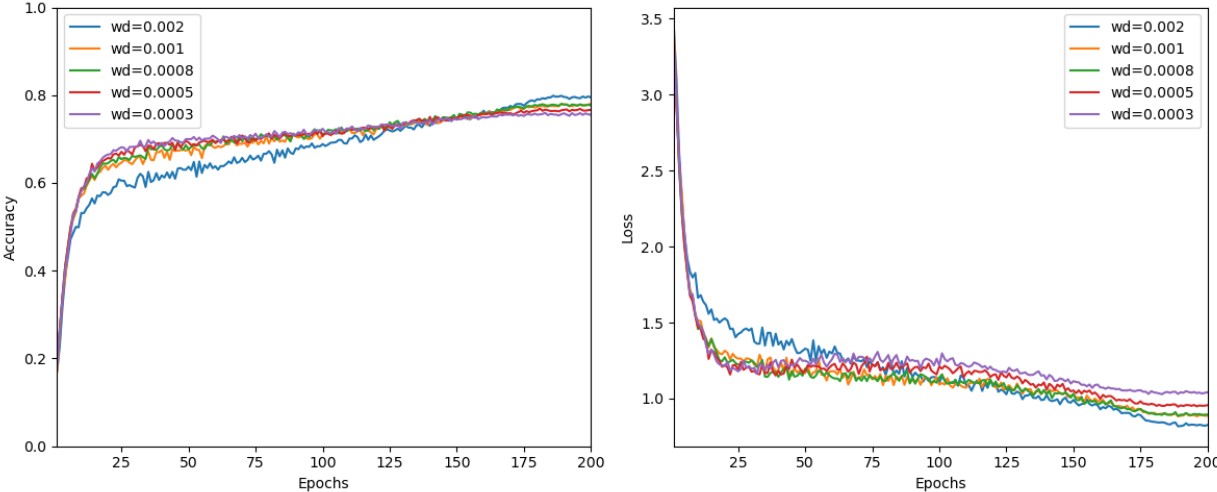

Figure 9: Comparison of accuracy and loss curves for different weight decay $\in$ $\{0.002, 0.001, 0.0008, 0.0005, 0.0003\}$ while rho is set to 0.05 and learning rate is set to 0.03.

aware methods. This stability reflects the core mechanism of LSAM, in which improving local integrability of the loss ($\mathscr{L}_S(\omega) \in L^p_{loc}$ with $p > d$) reduces high-frequency oscillations in the optimization process.

The impact of each hyperparameter matches our theoretical predictions. The value of $\rho$ determines the smoothing scale and controls the balance between bias and variance. The value of $k$ mainly affects the reliability of the estimated gradients and has little influence on the overall optimization path. Taken together, these results show that LSAM is not very sensitive to hyperparameter choices. The narrow variance bands across all experiments, even when hyperparameters vary widely, indicate that LSAM reliably moves toward regions with high Lebesgue regularity. This robustness appears not only in the final accuracy but also in the training behavior itself. The loss curves remain steady without large spikes, and the accuracy curves avoid sudden drops. This behavior directly reflects the link between local integrability and algorithmic stability established in Section 4.

## 6 Conclusion

This work introduced a new theoretical framework for understanding loss landscape flatness based on the Lebesgue differentiation theorem. We proved that Lebesgue points of the empirical loss correspond to parameters with low oscillation and strong generalization, and that local averaging convergence holds for them, without relying on Hessian or Lipschitz smoothness assumptions. Motivated by this insight, we proposed LSAM. A Monte Carlo approximation enables effective training.

Experiments on CIFAR-10 and CIFAR-100 demonstrate the practical impact of this framework. LSAM achieves the superior or comparable state-of-the-art accuracy and consistently delivers low and stable variance comparing the SGD and SAM series methods. Even on large capacity networks where SAM type methods occasionally lead in accuracy, LSAM remains competitive while maintaining superior stability. Hyperparameter studies further confirm our theoretical predictions that the smoothing radius controls the bias variance trade-off and that the number of perturbation samples influences estimator robustness without altering the optimization trajectory.

Overall, this work rethinks generalization as a consequence of almost everywhere regularity of the loss surface and shows that enforcing average case flatness is both theoretically principled and empirically effective. The combination of analytical insight and empirical performance indicates that incorporating measure-theoretic

concepts such as local integrability and averaging may offer a promising direction for future research on optimization and generalization in deep learning.

**Acknowledgments**

This work has been supported by the predoctoral program AGAUR-FI ajuts (2026 FI-3 00470) Joan Oró, which is backed by the Secretariat of Universities and Research of the Department of Research and Universities of the Generalitat of Catalonia, as well as the European Social Plus Fund, Beatriu de Pinós del Departament de Recerca i Universitats de la Generalitat de Catalunya (2022 BP 00256), Grant PID2021-128178OB-I00 funded by MCIN/AEI/10.13039/501100011033, ERDF "A way of making Europe".

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

## A  Proofs of Theoretical Results and Supplementary Experiments

**Theorem 1** (Generalization Bound via Local Average Loss). *Assume $l(\omega; z) \in [0, 1]$ for all $\omega, z$, and let $\rho > 0$. For any parameter $\omega \in \mathbb{R}^d$, the following deterministic decomposition holds:*

$$|Gen(\omega)| \leq \underbrace{\omega_{\mathscr{L}_D}(\rho)}_{\text{(I) population smoothness}} + \underbrace{\left|\overline{\mathscr{L}}_D^\rho(\omega) - \overline{\mathscr{L}}_S^\rho(\omega)\right|}_{\text{(II) sampling deviation}} + \underbrace{\mathbb{E}_{u \sim \text{Unif}(B_\rho)}\left[\left|\mathscr{L}_S(\omega + u) - \mathscr{L}_S(\omega)\right|\right]}_{\text{(III) smoothing bias / center-based local oscillation}} . \tag{38}$$

*Here*

$$\overline{\mathscr{L}}_S^\rho(\omega) = \frac{1}{|B_\rho|} \int_{B_\rho(\omega)} \mathscr{L}_S(\omega')\, d\omega', \qquad \overline{\mathscr{L}}_D^\rho(\omega) = \frac{1}{|B_\rho|} \int_{B_\rho(\omega)} \mathscr{L}_D(\omega')\, d\omega', \tag{39}$$

*and $\omega_f(\rho) := \sup_{\|\omega - \omega'\| \leq \rho} |f(\omega) - f(\omega')|$ denotes the modulus of continuity of $f$.*

*In particular, the generalization error at $\omega$ is governed by three components: (i) the local smoothness of the population loss ($\omega_{\mathscr{L}_D}(\rho)$); (ii) a sampling deviation term comparing local averages of the population and empirical losses; and (iii) the smoothing bias of the empirical loss, controlled by the average center-based local oscillation within the neighborhood $B_\rho(\omega)$.*

*Proof.* We decompose the generalization error and then bound each part.

**Step One: Decomposition of the generalization error.**

Recall

$$\mathscr{L}_D(\omega) = \mathbb{E}_{z \sim \mathcal{D}}[l(\omega; z)], \qquad \mathscr{L}_S(\omega) = \frac{1}{n} \sum_{i=1}^{n} l(\omega; z_i). \tag{40}$$

For any $\omega$,

$$\begin{aligned} Gen(\omega) &= \mathscr{L}_D(\omega) - \mathscr{L}_S(\omega) \\ &= \left(\mathscr{L}_D(\omega) - \overline{\mathscr{L}}_D^\rho(\omega)\right) + \left(\overline{\mathscr{L}}_D^\rho(\omega) - \overline{\mathscr{L}}_S^\rho(\omega)\right) + \left(\overline{\mathscr{L}}_S^\rho(\omega) - \mathscr{L}_S(\omega)\right). \end{aligned}$$

Taking absolute values and applying the triangle inequality,

$$|Gen(\omega)| \leq \underbrace{|\mathscr{L}_D(\omega) - \overline{\mathscr{L}}_D^\rho(\omega)|}_{\text{(I)}} + \underbrace{|\overline{\mathscr{L}}_D^\rho(\omega) - \overline{\mathscr{L}}_S^\rho(\omega)|}_{\text{(II)}} + \underbrace{|\overline{\mathscr{L}}_S^\rho(\omega) - \mathscr{L}_S(\omega)|}_{\text{(III)}}. \tag{41}$$

**Step Two: Control of the three terms.**

*Term (I): population smoothness.* By definition of $\overline{\mathscr{L}}_D^\rho(\omega)$,

$$\overline{\mathscr{L}}_D^\rho(\omega) = \mathbb{E}_{u \sim \text{Unif}(B_\rho)}\big[\mathscr{L}_D(\omega + u)\big]. \tag{42}$$

Hence

$$\begin{aligned}
|\mathscr{L}_D(\omega) - \overline{\mathscr{L}}_D^\rho(\omega)| &= \Big|\mathbb{E}_u\big[\mathscr{L}_D(\omega) - \mathscr{L}_D(\omega + u)\big]\Big| \\
&\leq \mathbb{E}_u\big[|\mathscr{L}_D(\omega) - \mathscr{L}_D(\omega + u)|\big] \\
&\leq \omega_{\mathscr{L}_D}(\rho),
\end{aligned} \tag{43}$$

since $\|u\| \leq \rho$ for $u \in B_\rho$.

*Term (II): sampling deviation.* The quantity

$$\overline{\mathscr{L}}_D^\rho(\omega) - \overline{\mathscr{L}}_S^\rho(\omega) \tag{44}$$

is the difference between the population and empirical averages of the locally smoothed loss at the same parameter $\omega$. We leave this term in the form

$$|\overline{\mathscr{L}}_D^\rho(\omega) - \overline{\mathscr{L}}_S^\rho(\omega)|, \tag{45}$$

which can later be controlled by standard concentration inequalities when $\omega$ is fixed, or by stability/complexity arguments when $\omega = \omega_S$ depends on $S$.

*Term (III): smoothing bias of the empirical loss.* By definition,

$$\overline{\mathscr{L}}_S^\rho(\omega) = \mathbb{E}_{u \sim \text{Unif}(B_\rho)}\big[\mathscr{L}_S(\omega + u)\big]. \tag{46}$$

Therefore,

$$\overline{\mathscr{L}}_S^\rho(\omega) - \mathscr{L}_S(\omega) = \mathbb{E}_u\big[\mathscr{L}_S(\omega + u) - \mathscr{L}_S(\omega)\big]. \tag{47}$$

By Jensen's inequality,

$$\begin{aligned}
\big|\overline{\mathscr{L}}_S^\rho(\omega) - \mathscr{L}_S(\omega)\big| &= \Big|\mathbb{E}_u\big[\mathscr{L}_S(\omega + u) - \mathscr{L}_S(\omega)\big]\Big| \\
&\leq \mathbb{E}_u\big[|\mathscr{L}_S(\omega + u) - \mathscr{L}_S(\omega)|\big].
\end{aligned} \tag{48}$$

Thus, Term (III) is controlled by the average center-based local oscillation of the empirical loss.

**Step Three: Conclusion.**

Combining the bounds for (I)–(III) in equation 41 yields equation 38, which completes the proof. $\qquad \square$

*Remark* 1 (Relation to local-average oscillation). The preceding decomposition controls the pointwise generalization gap through the smoothing bias

$$\big|\overline{\mathscr{L}}_S^\rho(\omega) - \mathscr{L}_S(\omega)\big|. \tag{49}$$

Under the additional local Hölder/Sobolev regularity condition, this smoothing bias and the mean oscillation around the local average are both controlled.

Specifically, suppose that around $\omega$ there exist constants $C_\omega > 0$ and $\alpha \in (0, 1]$ such that

$$|\mathscr{L}_S(\omega + u) - \mathscr{L}_S(\omega)| \leq C_\omega \|u\|^\alpha, \qquad u \in B_\rho. \tag{50}$$

Then

$$\left| \overline{\mathscr{L}}_S^\rho(\omega) - \mathscr{L}_S(\omega) \right| \le C_\omega \rho^\alpha. \tag{51}$$

Moreover, by the triangle inequality,

$$|\mathscr{L}_S(\omega + u) - \overline{\mathscr{L}}_S^\rho(\omega)| \le |\mathscr{L}_S(\omega + u) - \mathscr{L}_S(\omega)| + |\mathscr{L}_S(\omega) - \overline{\mathscr{L}}_S^\rho(\omega)|. \tag{52}$$

Taking expectation over $u$ gives

$$\mathbb{E}_u\left[ |\mathscr{L}_S(\omega + u) - \overline{\mathscr{L}}_S^\rho(\omega)| \right] \le 2C_\omega \rho^\alpha. \tag{53}$$

When the Hölder regularity is induced by Sobolev embedding, one may take $\alpha = 1 - d/p$ under the condition $\mathscr{L}_S \in W_{\text{loc}}^{1,p}(\mathbb{R}^d)$ with $p > d$.

**Theorem 2** (Generalization from Sobolev Regularity and Integrability). *Suppose the empirical loss $L_S : \mathbb{R}^d \to [0,1]$ satisfies*

$$L_S \in W_{\text{loc}}^{1,p}(\mathbb{R}^d) \quad \text{for some } p > d. \tag{54}$$

*Equivalently, $L_S \in L_{\text{loc}}^p(\mathbb{R}^d)$ and $\nabla L_S \in L_{\text{loc}}^p(\mathbb{R}^d)$. Let $\alpha = 1 - d/p \in (0,1)$ be the Morrey exponent.*

*Then there exists a full-measure subset $\Omega \subset \mathbb{R}^d$ (i.e. $|\mathbb{R}^d \setminus \Omega| = 0$) such that for every $\omega \in \Omega$ there exist constants $C_\omega > 0$ and $\rho_\omega > 0$ with*

$$\sup_{\|u\| \le \rho} \left| L_S(\omega + u) - L_S(\omega) \right| \le C_\omega \rho^\alpha, \qquad 0 < \rho \le \rho_\omega. \tag{55}$$

*In particular, $L_S$ is locally Hölder continuous of order $\alpha$ at almost every Lebesgue point $\omega$.*

*Furthermore, let $\omega_S \in \Omega$ be the solution returned by the optimizer. Assume that the population loss $L_D$ is also locally Hölder continuous of order $\alpha$ near $\omega_S$, i.e., there exist $C_D > 0$ and $\rho_D > 0$ such that*

$$\sup_{\|u\| \le \rho} \left| L_D(\omega_S + u) - L_D(\omega_S) \right| \le C_D \rho^\alpha, \qquad 0 < \rho \le \rho_D. \tag{56}$$

*Then for any $\delta \in (0,1)$ and any $0 < \rho \le \rho_0 := \min\{\rho_\omega, \rho_D\}$, with probability at least $1 - \delta$ (over $S \sim \mathcal{D}^n$), the generalization error satisfies*

$$Gen(\omega_S) \le (C_D + C_S) \rho^\alpha + \sqrt{\frac{2\log(2/\delta)}{n}}, \tag{57}$$

*where $C_S := C_{\omega_S}$ is the Hölder constant of $L_S$ at $\omega_S$. Choosing, for example, $\rho \asymp n^{-1/(2\alpha)}$ yields an explicit generalization rate.*

*Proof.* The proof consists of two steps:

**(1) Local Hölder continuity follows from local integrability and Sobolev regularity.**

Since $L_S$ is bounded, we already have $L_S \in L_{\text{loc}}^1(\mathbb{R}^d)$. By assumption,

$$L_S \in W_{\text{loc}}^{1,p}(\mathbb{R}^d), \qquad p > d, \tag{58}$$

so both $L_S$ and its weak gradient $\nabla L_S$ are locally Lebesgue integrable of order $p$.

Fix $\omega \in \mathbb{R}^d$ such that $L_S$ admits a Lebesgue representative and $\nabla L_S$ is defined in the Sobolev sense at $\omega$; this holds for almost every $\omega$ by standard Sobolev theory. Since the ambient domain is $\mathbb{R}^d$, we may choose $R > 0$ such that the closed ball $\overline{B_R(\omega)}$ is compact, and hence

$$L_S \in W^{1,p}(B_R(\omega)), \qquad p > d. \tag{59}$$

By Morrey's inequality, there exists a Hölder-continuous representative of $L_S$ (still denoted $L_S$) and a constant

$$C = C\big(d, p, \|L_S\|_{W^{1,p}(B_R(\omega))}\big) > 0 \tag{60}$$

such that

$$|L_S(\omega_1) - L_S(\omega_2)| \leq C \|\omega_1 - \omega_2\|^\alpha, \qquad \forall\, \omega_1, \omega_2 \in B_{R/2}(\omega). \tag{61}$$

Setting $\rho_\omega := R/2$ and $C_\omega := C$, we obtain equation 55 for all $0 < \rho \leq \rho_\omega$. This proves that $L_S$ is locally Hölder continuous of order $\alpha$ at almost every point, with constants depending only on $d, p$ and the local Sobolev norm $\|L_S\|_{W^{1,p}(B_R(\omega))}$.

**(2) Substitute this regularity into the generalized boundary.**

We now combine the local Hölder flatness equation 55, equation 56 with the generalization decomposition of Theorem 1. For completeness, recall that for any $\omega$,

$$Gen(\omega) = \left(L_D(\omega) - \overline{L}_D^\rho(\omega)\right) + \left(\overline{L}_D^\rho(\omega) - \overline{L}_S^\rho(\omega)\right) + \left(\overline{L}_S^\rho(\omega) - L_S(\omega)\right), \tag{62}$$

where

$$\overline{L}_S^\rho(\omega) = \frac{1}{|B_\rho|} \int_{B_\rho(\omega)} L_S(\omega')\, d\omega', \qquad \overline{L}_D^\rho(\omega) = \frac{1}{|B_\rho|} \int_{B_\rho(\omega)} L_D(\omega')\, d\omega'. \tag{63}$$

We control the three terms at $\omega = \omega_S$ separately.

*Term I: Population smoothing error.* By the Hölder continuity of $L_D$ near $\omega_S$,

$$\left|L_D(\omega_S) - \overline{L}_D^\rho(\omega_S)\right| = \left| \frac{1}{|B_\rho|} \int_{B_\rho} \left(L_D(\omega_S) - L_D(\omega_S + u)\right) du \right| \leq C_D \rho^\alpha, \tag{64}$$

for all $0 < \rho \leq \rho_0$.

*Term II: Sampling deviation.* As in Theorem 1, $L_S(\omega) \in [0,1]$ implies that changing one sample modifies $\overline{L}_S^\rho(\omega_S)$ by at most $1/n$. By McDiarmid's bounded differences inequality, for any $\delta \in (0,1)$, with probability at least $1 - \delta$,

$$\left|\overline{L}_D^\rho(\omega_S) - \overline{L}_S^\rho(\omega_S)\right| \leq \sqrt{\frac{2\log(2/\delta)}{n}}. \tag{65}$$

*Term III: Sobolev-induced flatness of the empirical loss.* Using the Hölder continuity of $L_S$ at $\omega_S$, we have for all $u$ with $\|u\| \leq \rho_0$,

$$|L_S(\omega_S + u) - L_S(\omega_S)| \leq C_S \|u\|^\alpha. \tag{66}$$

Therefore,

$$\begin{aligned} \left|\overline{L}_S^\rho(\omega_S) - L_S(\omega_S)\right| &= \left| \frac{1}{|B_\rho|} \int_{B_\rho} \left(L_S(\omega_S + u) - L_S(\omega_S)\right) du \right| \\ &\leq \frac{C_S}{|B_\rho|} \int_{B_\rho} \|u\|^\alpha\, du \leq C_S \rho^\alpha, \end{aligned} \tag{67}$$

for all $0 < \rho \leq \rho_0$, after absorbing the geometric factor into the constant.

*Conclusion.* Combining the three bounds and evaluating at $\omega = \omega_S$, we obtain

$$Gen(\omega_S) \leq (C_D + C_S)\rho^\alpha + \sqrt{\frac{2\log(2/\delta)}{n}}, \tag{68}$$

which is equation 57. Choosing $\rho$ as a function of $n$, for example $\rho \asymp n^{-1/(2\alpha)}$, yields the stated generalization rate. $\qquad\square$

Figure 10 (full training period) illustrates the entire training process for all models. The fast convergence during the first 25 epochs is typical for optimization in high-dimensional non-convex problems, where the local average loss still provides useful gradient information to move away from sharp minima. Although all models reach similar final accuracy (around 96 to 97 percent), their convergence speeds differ.

From the viewpoint of local integrability (Theorem 2), this early phase suggests that the optimizer is moving toward regions where $\mathscr{L}_S$ belongs to $L_{lpc}^P$ with $p > d$. The fact that all models show rapid progress indicates

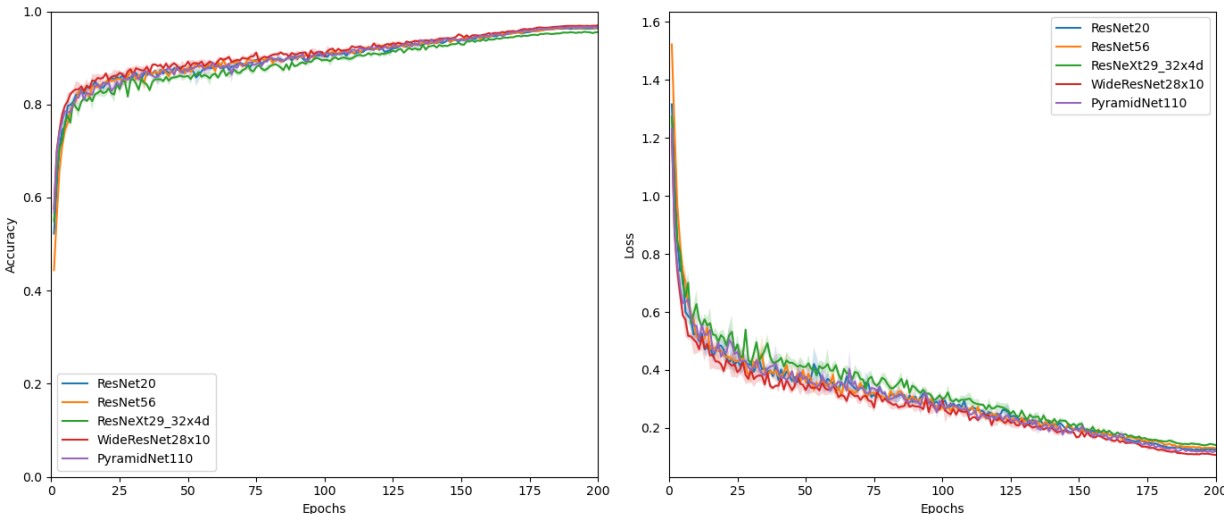

Figure 10: Comparison of test accuracy (left) and loss curves (right) across models throughout 200 training epochs for CIFAR-10.

that the CIFAR-10 loss landscape contains many such regions. WideResNet28×10 and PyramidNet110 converge slightly faster, which suggests that their architectures create loss surfaces with higher local regularity due to their larger capacity and natural smoothing effects.

The loss curves also show a second observation. The steady decrease in loss after the initial phase suggests that the local average loss used by LSAM helps the optimizer avoid regions with irregular oscillations that would appear as sudden jumps in the loss. This matches our theoretical result that minimizing $\overline{\mathscr{L}}_S$ reduces high-frequency variations in the loss surface and keeps the optimization in regions where $\mathscr{L}_S$ is Hölder continuous with exponent $\alpha > 0$.

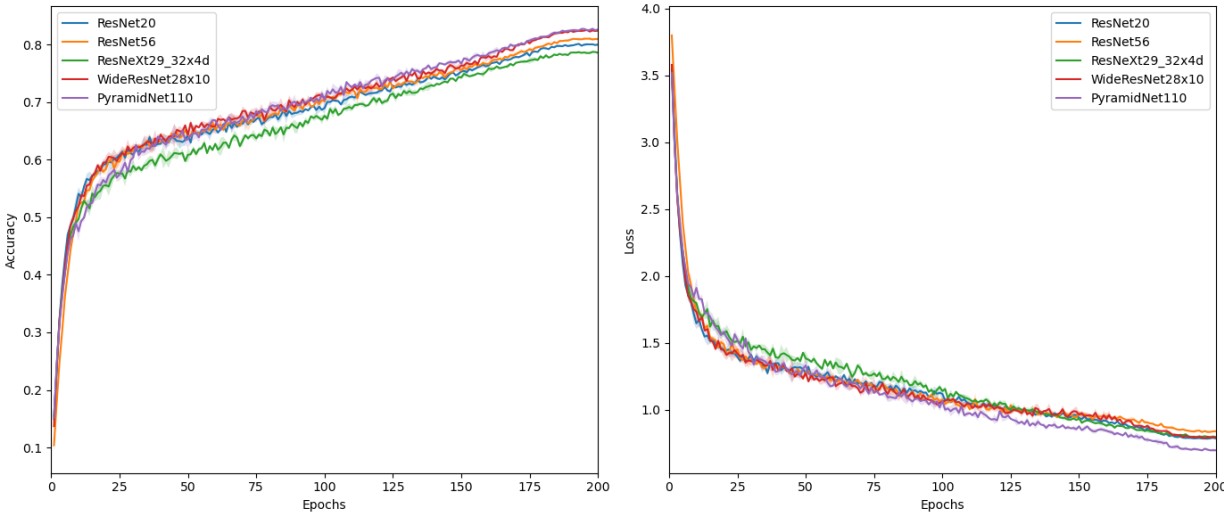

Figure 11: Comparison of test accuracy (left) and loss curves (right) across models throughout 200 training epochs for CIFAR-100.

Figure 11 shows a natural performance drop compared with CIFAR-10. The final accuracy falls in the range of 78% to 82%, rather than the 96% to 97% seen on CIFAR-10. The slower convergence and higher final loss reflect the greater difficulty of the CIFAR-100 dataset, which reduces the degree of local integrability that the loss function can achieve during training.

Crucially, the differences between models become more apparent in this setting. WideResNet28×10 and PyramidNet110 maintain a clear advantage over the other architectures throughout training, which suggests that these models preserve local regularity more effectively on the more challenging CIFAR-100 dataset. According to Theorem 2, complex datasets impose stronger requirements on ensuring that $\mathscr{L}_S$ belongs to $L^p_{loc}$ with $p > d$, and models that can maintain this property tend to generalize better.

The loss curves also show that as dataset complexity increases, the benefit of optimizing for local integrability becomes more important. While all models behave similarly during the early stage of training (0 to 25 epochs), clear differences appear after about 50 epochs. This marks the period when LSAM begins to strongly affect the optimization path by guiding the search toward flatter and more regular regions of the loss landscape. This tendency is most evident in WideResNet28×10 and PyramidNet110, which keep a consistent advantage in loss throughout the entire training process.

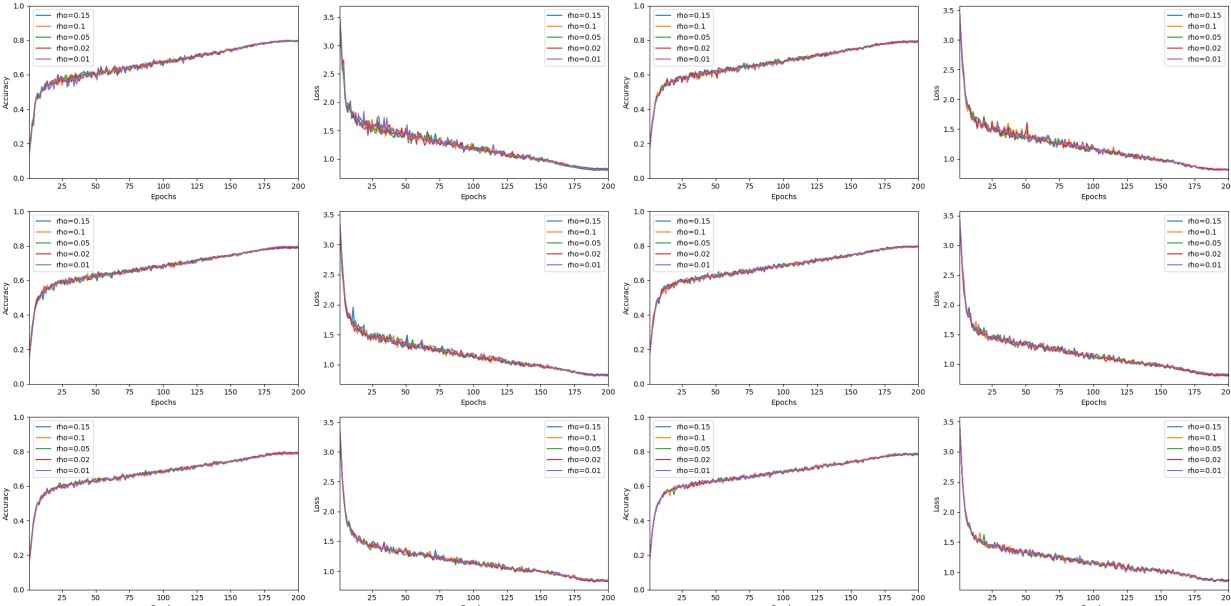

Figure 12: Accuracy and loss curves for different values of $k \in \{1, 2, 3, 4, 8, 16\}$ (arranged from top to bottom, left to right). Within each subplot pair, curves correspond to different values of $\rho \in \{0.15, 0.10, 0.05, 0.02, 0.01\}$ during 200 epochs.

Figures 12 and 4 show the training behavior of LSAM under different smoothing radii $\rho \in \{0.01, 0.02, 0.05, 0.1, 0.15\}$. The full training curves in Figure 12 reveal several patterns that agree with our theoretical analysis.

During the early part of training (epochs 0 to 50), all values of $\rho$ show similar convergence speed. This indicates that the smoothing operation does not slow down the initial optimization. This matches Theorem 2 in Section 4, which states that the bias introduced by local averaging is controlled by $\rho^\alpha$ with $\alpha = 1 - d/p$ greater than zero. For small values of $\rho$, such as 0.01 to 0.05, this bias is very small at the beginning of training, which allows the algorithm to adapt well to the changes in the loss.

In the middle part of training (epochs 50 to 150), the differences between the choices of $\rho$ become clear. The settings $\rho = 0.05$ and $\rho = 0.02$ consistently show better accuracy and lower loss compared to $\rho = 0.01$ and the larger values $\rho = 0.1$ and $\rho = 0.15$. This supports our prediction that there is an optimal smoothing

radius that balances bias and variance. When $\rho$ is too small, such as 0.01, the local mean $\mathscr{L}_S^\rho$ does not provide enough regularization and the optimizer may settle near sharp minima that generalize poorly. When $\rho$ is too large, such as 0.15, the bias term $\rho^\alpha$ becomes large and the function being optimized no longer reflects the true empirical loss.

In the last part of training (epochs 150 to 200, Figure 4), the differences in final performance become even clearer. The patterns in the curves provide direct support for our idea that better local integrability of the loss leads to stronger generalization.

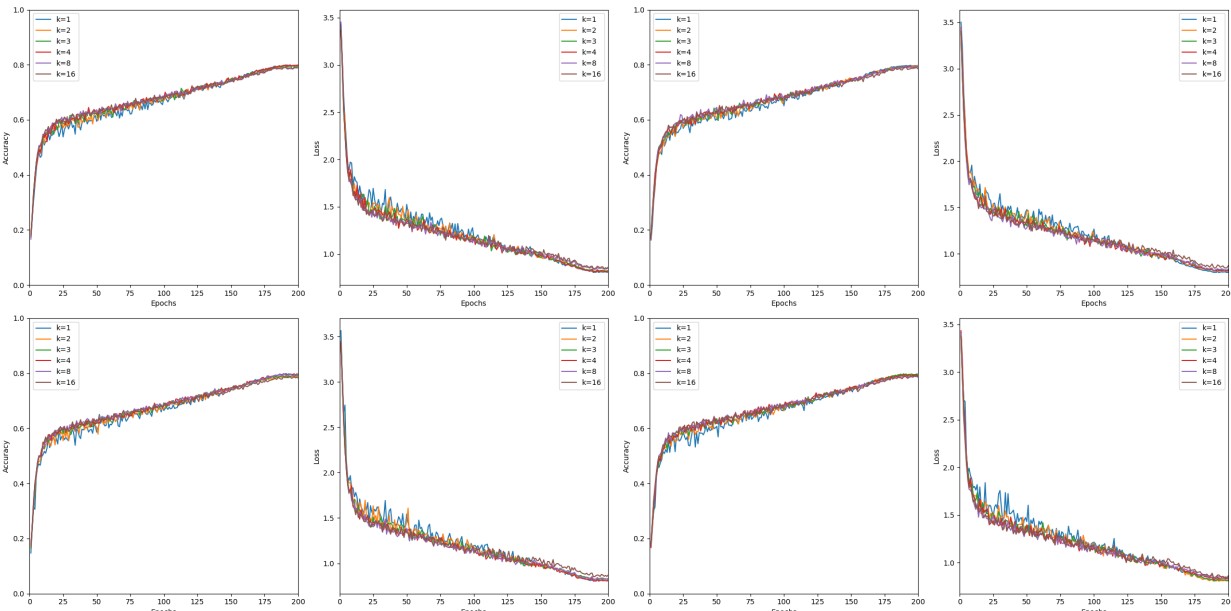

Figure 13: Accuracy and loss curves for different values of $\rho \in \{0.10, 0.05, 0.02, 0.01\}$ (arranged from top to bottom, left to right). Within each subplot pair, curves correspond to different values of $k \in \{1, 2, 3, 4, 8, 16\}$ during 200 epochs.

Figures 13 and 5 show how the parameter $k$ affects the training process. Figure 13 supports an important prediction of our theoretical framework. After a certain point, increasing $k$ gives only small improvements. All choices of $k$ reach almost the same final accuracy of about 0.80. This confirms our earlier statement that $k \geq 1$ is sufficient for effective local averaging, while using values of $k$ larger than 8 does not provide clear benefits.

The training curves show that increasing $k$ mainly reduces the variance of the gradient estimates rather than changing the direction of the optimization path. This can be seen from the smoother loss curves for larger $k$. For example, the curves for $k = 16$ and $k = 1$ reach similar results in the middle stage of training (epochs 50 to 150), but the fluctuations for $k = 16$ are much smaller. This supports the variance analysis discussed earlier.

