# OpenReview forum: "Sharpness-Aware Minimization Driven by Local-Integrability Flatness"
_TMLR — Decision pending for TMLR_

### Review · Reviewer_TTxK · 2026-03-28

**Summary Of Contributions:**

The authors propose Lebesgue Sharpness-Aware Minimization (LSAM), a measure-theoretic method for optimizing the loss landscape under local paramter perturbations. Classical gradient descent such as SGD yields local minima, but have no regard for the landscape around the found minima, resulting in high volatility against perturbations and poor generalization. The standard practice is to employ sharpness-aware minimization (SAM) in order to bias training towards "flatter" regional minima, and experimentally has shown its superiority over SGD and other methods. However, the authors argue that SAM is too pessimistic of a method, prone to outlier influence, and relies on smoothness assumptions that are not always satisfied in modern network architectures.

Based on measure-theoretic results such as Lebesgue differentiation theorem and Sobolev regularity, LSAM focuses on a "beyond worst-case" interpretation of the local loss landscape, favoring average loss within a local neighborhood as opposed to the worst-case loss gradient. The minimizer is implemented with a Monte-Carlo sampler to estimate the local loss average as a practical realization of the theoretical model. Experiments are set up and conducted on CIFAR-10 and CIFAR-100 image classification benchmarks against SGD and other iterations of SAM. LSAM obtains an impressive improvement on smaller models on both CIFAR-10 and -100, but interestingly falls short with larger models against classical SAM. The authors conject that the loss landscape of larger models are already smooth, favoring worst-case sharpness over average-case sharpness.

**Audience:**

Yes

**Audience Explanation:**

Sharpness-aware minimization is a very well-motivated and well-studied topic in the field of deep learning recently, evident in the vast amount of architectures and alternatives to SAM in literature. Combined with recent interests in beyond worst-case theoretical viewpoints in fields of machine learning and statistics as a whole, it is natural to arrive at a starting such as the paper's proposed more optimistic view of the loss landscape in gradient descent.

The motivation behind this research direction can be strengthened by explicit citation of prior work raising the question and concern about the inherent pessimism of SAM and other worst-case-styled objectives.

**Broader Impact Concerns:**

N/A.

**Claims And Evidence:**

Yes

**Claims Explanation:**

The paper is overall well-structured with clear writing, and the theoretical framework is believable at the first glance, with a quick check yielding no obvious errors and discrepancies (although I did not have time to scrutinize every claim and lemma). Intuitively, an average-case or beyond worst-case analysis of the loss landscape will more consistently yield better generalization with real-world networks, although care should be taken to analyze (perhaps from an empirical standpoint) extreme cases where the average-case analytical framework might break -- such as the experimental results presented on larger models such as PyramidNet-110.

I am somewhat taken by surprise at the experimental comparison between LSAM and SAM on larger models, however, since the authors did not elaborate on this part of their findings in prior introductory sections. Throughout the introduction I find myself under the impression that LSAM is consistently a better-performing minimizer than SAM on all fronts. I encourage the authors to take into consideration this facet of their empirical results and include a more thorough introduction of their experimental results in the first few pages and sections.

**Requested Changes:**

As outlined above, although the submission is in good shape, I would like to request (1) a more thorough, up-front discussion of LSAM's performance degradation on larger models, and perferably (2) explicit citations of prior works raising questions that their work can answer.

The paper also contains an amount of typos, such as:
- page 4, "SAM and Variants", line 4: "...for each parameter iterate \omega, ...";
- page 5, "Methodology", line 2 after eq (3): Additionally, if $\mathcal{L}_S \in L_{lpc}^P(\mathbb{R}^d)$ for some $p > d$...", the "lpc" in the subscript is a typo, and $p$ (lower-case) is not present in the formula -- perhaps meant for $P$ (upper-case).
- page 14, "Comparison Algorithms and Evaluation Metrics" and "The main model used for the experiment" should surely be distinct paragraph-sections.
- page 15, line 3: "However, performance trade-offs in high-capacity models." Personally I've never seen trade-off been used as a verb, but perhaps it is reasonable.

I think the submission can very much benefit from a more thorough grammatical check.

---

> ### Author Response · Authors · 2026-05-06
> **Response to Reviewer TTxK**
>
> We thank the reviewer for the careful and constructive review. We appreciate the positive assessment of the paper's structure, motivation, theoretical framework, and empirical evidence. We also thank the reviewer for pointing out that the performance trade-off on larger models should be discussed more explicitly and earlier in the paper. We respond to the requested changes below.
>
> ### Discussion of LSAM on larger models
>
> We thank the reviewer for highlighting this point. We agree that the introduction should not give the impression that LSAM uniformly dominates SAM-type methods in every architecture and dataset setting. Our empirical results show a more nuanced picture. LSAM provides substantial improvements on small and medium models, such as ResNet-20 and ResNet-56, especially on CIFAR-100. However, on larger or higher-capacity models such as ResNeXt-29-32x4d and PyramidNet-110, LSAM can be competitive but does not always outperform SAM or SSAM.
>
> This behavior is consistent with our interpretation of LSAM as an average-case local smoothing method. In lower-capacity models, the empirical loss landscape can be more irregular, and average-based smoothing helps avoid sharp or noisy regions that generalize poorly. In contrast, high-capacity architectures may already induce smoother local geometry or may benefit from preserving certain sharp but useful local structures. In these cases, the worst-case perturbation used by SAM-type methods may align more closely with the generalization behavior of the model, while LSAM's averaging may smooth out some informative local variations.
>
> We will revise the introduction to emphasize that the main advantage of LSAM is theoretical as well as algorithmic. Starting from the Lebesgue differentiation theorem, LSAM replaces worst-case sharpness with a local-average objective and yields a generalization bound based on a local modulus of continuity and a Sobolev-induced flatness term, without relying on global Lipschitz assumptions, Hessian smoothness, or second-order information. This makes the framework more suitable for modern nonsmooth neural networks, while the empirical results in Section 5 further show when this average-case formulation is most beneficial.
>
> ### Relation to prior work on the pessimism of worst-case objectives
>
> We thank the reviewer for the suggestion. In the current manuscript, we already discuss SAM and several variants, including ASAM, GASAM, and SSAM. We will further strengthen this discussion by more explicitly connecting LSAM to prior works that point out limitations of max-based or worst-case sharpness objectives, such as sensitivity to the perturbation radius, conservative behavior, and instability in certain optimization regimes [andriushchenko2022towards; bartlett2023dynamics; tan2025stable].
>
> Starting from the Lebesgue differentiation theorem, LSAM replaces worst-case sharpness with a local-average objective:
>
> $$
> \max_{\|u\|\leq \rho} \overline{\mathscr{L}}_S(\omega+u)
> \quad
> \longrightarrow
> \quad
> \mathbb{E}
> \left[
> \overline{\mathscr{L}}_S(\omega+u)
> \right],
> \qquad
> u \sim \mathrm{Unif}(B(0,\rho)).
> $$
>
> This reformulation is not only less sensitive to isolated steep or noisy perturbation directions, but also yields a generalization bound based on a local modulus of continuity and a Sobolev-induced flatness term around the learned parameter. Unlike analyses relying on global Lipschitz assumptions, Hessian smoothness, or second-order information, our framework is more suitable for modern nonsmooth neural networks. The empirical results in Section 5 further illustrate the settings in which this average-case formulation is most beneficial.
>
> ### Typos and grammatical issues
>
> We thank the reviewer for listing these issues. We will carefully proofread the manuscript and correct the identified typos and grammatical problems.

---

### Review · Reviewer_aa6A · 2026-04-01

**Summary Of Contributions:**

This paper proposes Lebesgue Sharpness-Aware Minimization (LSAM) for accounting the neural network optimization under parameter perturbations in measure theoretic arguments. To my understanding of the two core arguments, the Lebesgue differentiation theorem is a justification for utilizing local averages instead of relying on the worst-case, and local Sobolev regularity accounts for Hölder-flatness that is used to bound each algorithmic perturbation. This approach can be useful since it connects real analytic arguments into the measure theory, and the Sobolev regularity enables more finer analysis using general conditions i.e., concentration of measures. Based on this theory, the authors proposed the LSAM step, a straightforward algorithm from perturbation from tractable distributions, and showed certain performance improvement on many standard neural nets compared to other benchmarks other than SAM.

**Audience:**

Yes

**Audience Explanation:**

SAM are in a great interest in the community who are working on loss landscapes, or learning theory in general.

**Claims And Evidence:**

Yes

**Claims Explanation:**

The authors overall have provided accurate and convincing results both in theory and applications. I think most of the theoretical arguments are sound (I have not checked the entire step line-by-line). Although the performance gains are not stellar with overlapping performance regions with original SAM in tables, the paper is faithful for providing various aspects of the algorithm LSAM by comparative studies.

**Requested Changes:**

I am not an expert on this topic; Please provide justifications if the changes are not applicable.

* The experiments cover CIFAR-10 and 100 for comparing complexity of datasets. I am curious whether does this work predict some of complexity for the models, or it is intentionally designed to be agnostic like PAC-Bayes? Does LSAM behave differently than SAM if the models become complex (related to the conclusion)?
* Section 3 (Methodology) is somewhat terse and less clear compared to other parts. The authors are encouraged to elaborate on computational motivations and the notion of unbiasedness. It would be better there is a toy example, or illustrative figure of the procedure.
* Please engage more or discussion on computational limitations and why this reformulation can have potential to be significant in Section 6.

---

> ### Author Response · Authors · 2026-05-06
> **Response to Reviewer aa6A**
>
> We thank the reviewer for the careful and constructive assessment of our work. We appreciate the recognition that LSAM provides a measure-theoretic alternative to worst-case sharpness and that the combination of Lebesgue differentiation and local Sobolev regularity gives a useful analytical framework for studying local flatness and generalization. We address the requested changes below.
>
> ### Model complexity and behavior relative to SAM
>
> We thank the reviewer for raising this important point. LSAM is not designed to predict model complexity in the same sense as a capacity measure or a PAC-Bayes complexity term. Rather, our framework is primarily local and algorithmic: it characterizes how the empirical loss behaves in a neighborhood of the learned parameter through local averaging, local oscillation, and Sobolev-type regularity. In this sense, LSAM is closer in spirit to a local flatness-based optimization principle than to a global model-complexity measure.
>
> That said, the behavior of LSAM can indeed depend on the effective complexity and capacity of the architecture. Our experiments suggest that LSAM is especially effective for small and medium models, where the loss landscape is more irregular and the average-case smoothing effect helps avoid sharp or noisy regions. This can be seen in the strong improvements on ResNet-20 and ResNet-56 for both CIFAR-10 and CIFAR-100. For larger models such as ResNeXt-29-32x4d and PyramidNet-110, the advantage of LSAM becomes more nuanced: these architectures may already induce smoother optimization landscapes, and worst-case sharpness methods such as SAM or SSAM can sometimes perform competitively or slightly better. This observation is consistent with our discussion in Section 5.2, where we note that average-based smoothing may sometimes smooth out fine but useful structures in high-capacity models.
>
> Therefore, our interpretation is that LSAM is model-complexity aware in an empirical and local-geometry sense, but not intended as a standalone model-complexity estimator. We will clarify this point in the conclusion and discussion: LSAM is agnostic to architecture at the algorithmic level, but its practical gains depend on how the model architecture shapes the local regularity of the loss surface.
>
> ### Clarification of Section 3 and computational motivation
>
> We thank the reviewer for pointing this out. The purpose of Section 3 is to translate the local-average objective
>
> $$
> \overline{ \mathscr{L}}_S^\rho( \omega) =
> \mathbb{E}
> \left[
> \overline{\mathscr{L}}_S( \omega+u)
> \right],
> \qquad
> u \sim \mathrm{Unif}(B(0,\rho))
> $$
>
> into a practical optimization algorithm. Since the exact integral over the perturbation ball is intractable in high-dimensional neural networks, LSAM uses a Monte Carlo approximation. With perturbations $u_1,\ldots,u_k$ sampled independently from $B(0,\rho)$, the gradient estimator is
>
> $$
> \bar{g}=
> \frac{1}{k}\sum_{i=1}^k
> \nabla_\omega \mathscr{L}_S(\omega+u_i).
> $$
>
> This estimator is unbiased for the gradient of the local average loss under the standard interchange of gradient and expectation:
>
> $$
> \mathbb{E}
> \left[
> \bar{g}
> \right]=
> \nabla_\omega \mathscr{L}_S^\rho(\omega).
> $$
>
> Thus, LSAM is not heuristically averaging arbitrary perturbed gradients; it is performing stochastic optimization on a well-defined smoothed objective.
>
> ### Computational limitations and significance of the reformulation
>
> We agree that the computational limitations and significance of LSAM deserve more explicit discussion. LSAM introduces a computational overhead proportional to the number of Monte Carlo samples $k$, since each update requires evaluating gradients at $k$ perturbed parameter points. This is the main computational cost of the method. However, this cost is also controllable: $k$ directly trades off estimator variance and computation. In practice, the method does not require Hessian computation, eigenvalue estimation, or second-order information, which keeps the implementation simple and compatible with modern nonsmooth architectures such as ReLU networks.
>
> The significance of the reformulation is that it replaces the max-based local objective of SAM with an expectation-based local objective:
>
> $$
> \max_{\|u\|\leq \rho}\overline{\mathscr{L}}_S(\omega+u)
> \quad
> \text{versus}
> \quad
> \mathbb{E}
> \left[
> \overline{\mathscr{L}}_S(\omega+u)
> \right],
> \qquad
> u \sim \mathrm{Unif}(B(0,\rho)).
> $$
>
> This change is not merely algorithmic. It also changes the theoretical object being optimized. The local average loss is directly connected to the Lebesgue differentiation theorem and to the local oscillation term in our generalization decomposition. Under Sobolev regularity, this local oscillation can be quantitatively controlled through a Hölder-type bound. Therefore, LSAM provides a direct bridge between a computable training algorithm and a measure-theoretic notion of local flatness.
>
> We will update Section 3 to improve the exposition of the LSAM procedure.

---

### Review · Reviewer_AkMq · 2026-05-03

**Summary Of Contributions:**

The paper proposes Lebesgue Sharpness-Aware Minimization (LSAM), a method that can locally smoothen the loss landscape. The paper first introduces the algorithm, then analyzes the generalization bound, then shows experiments that i) compare the performance of LSAM with other methods, as well as ii) exhibits the effect of different (hyper)parameters in LSAM.

**Additional Comments:**

N.A.

**Audience:**

Yes

**Audience Explanation:**

Smoothing the loss landscape  to improve training dynamics and sharpness-aware methods have long been an essential and interesting topic in machine learning, as far as I know.

**Broader Impact Concerns:**

N.A.

**Claims And Evidence:**

No

**Claims Explanation:**

1. Section 4 is presented in an extremely unclear way (see requested changes below).

2. Many figures in Section 5 are crowded with lines of different colors, making it difficult to observe the results claimed by the authors.

3. Experiment settings is unclear (see requested changes below).

4. Section 5.6 seems to discuss the effect of learning rate and weight decay on different (fixed) radius, instead of the effect of perturbation radius as indicated in the subtitle of this section.

5. "smoothing radius" and "perturbation radius" are used interchangeably throughout Section 5. The authors should stick to one term. I suggest changing the "smoothing radius" because it only occurs three times in Section 5.

6. Other things (see requested changes below).

**Requested Changes:**

1. Reorganize Section 4. State the theorems/corollaries/... clearly, instead of mixing the results with the proofs and discussions. Then present the proofs for the theorems/corollaries/... Rewrite the things in "[]", e.g. "[Generalization Bound via Local Average Loss]" in a similar way, or simply delete these things.

2. Delete repetitive formulas. For example, equation (19) and (20).

3. Explain how the term "$\bar{\mathcal{L}_S}^\rho (\omega)$" is obtained. Sorry I do not understand; what I can obtain is simply $\mathbb{E}_u[\mathcal{L}_S(\omega + u) - \mathcal{L}_S(\omega)] \le \mathbb{E}_u [|\mathcal{L}_S(\omega + u) - \mathcal{L}_S(\omega)|]$".

4. State explicitly your Theorem 4.1 and Theorem 4.2, and "Theorem 2" as mentioned on e.g. page 6, 14, 16, 17, 18, etc.

5. Define $L^P_{loc}(\mathbb{R}^d)$ on page 5.

6. On page 6, "... Monte Carlo estimator $\bar{g}$" should be "... Monte Carlo estimator $\bar{\textbf{g}}$".

7. On page 6, "LSAM minimizes $...[\mathcal{L}_{S} (\omega + u)$" should be "LSAM minimizes $...[\mathcal{L}_S(\omega + u) \textbf{]}$" -- right square brack is missing.

8. On page 8, it is better to use other notations than f when defining $\omega_f(\rho)$, because the model is denoted by $f$.

9. On Page 11, change all the "$L_D$" and "$L_S$" to "$\mathcal{L}_D$" and "$\mathcal{L}_S$", respectively.

10. Please specify the models and loss functions used in the experiments in Section 5.

11. On page 18, explain the meaning of bias in "...the bias introduced by local averaging ..."

12. In Section 5.4 and Section 5.5, explain the meaning of "K". Do you mean the number of samples $k$ (for computing the local average of loss) in Algorithm 1?

13. Re-plot the figures in Section 5.4, or explain how to read off the results from the current figures. The figures look almost identical. For example, it is unclear to me how the figures show that in the early part of training, "the bias introduced by local averaging is controlled by $\rho^\alpha$" (note that bias and model are unclear either) , or that "$\rho = 0.05$ and $\rho = 0.02$ show better accuracy and lower loss compared to $\rho = 0.01$ and the larger values $\rho = 0.1$ and $\rho = 0.15$".

14. Rewrite the subtitle or the content of Section 5.6, as this section does not seem to discuss the effect of perturbation radius, but the effect of learning rate and weight decay for different (fixed) perturbation radius.

15.  Re-plot the figures in Section 5.6, or explain how to read off the results from the current figures. For example, it is unclear to me how the figures show that in Figure 13 "the right panel indicates that all weight decay settings lead to smooth loss curves".

---

> ### Author Response · Authors · 2026-05-06
> **Response to Reviewer AkMq**
>
> We thank the reviewer for reading and comments. We are grateful for these suggestions and will revise accordingly
>
> ### Clarity and organization of Section 4
>
> We thank the reviewer for the suggestion. Section 4 was written step by step so that the decomposition and proof can be checked line by line. Equations (19) and (20) are not repetitive: they present two equivalent directions of the same local-averaging identity, useful for the theorem proof and derivation. Keeping both formulas makes the transition from local average loss to local oscillation explicit. Rather than deleting them, we will revise the text to better explain their role.
>
> We agree that the presentation can be easier to parse. In the revision, we will more clearly label theorem statements, proof steps, and explanatory remarks in Section 4. We will also consistently refer to Theorem 4.1 and Theorem 4.2 throughout the paper. These changes will improve readability while preserving the theoretical derivation.
>
> ### Clarification of the term $\overline{\mathscr{L}}_S^\rho(\omega)$
>
> $\overline{\mathscr{L}}_S^\rho(\omega)$ denotes the local average of the empirical loss around $\omega$:
>
> $$
> \overline{\mathscr{L}}_S^\rho(\omega)=
> \frac{1}{|B(0,\rho)|}
> \int
> \mathscr{L}_S(\omega')\,d\omega',
> \qquad
> \omega' \in B(\omega,\rho).
> $$
>
> Equivalently, with $u \sim \mathrm{Unif}(B(0,\rho))$,
>
> $$
> \mathscr{L}_S^\rho(\omega)=
> \mathbb{E}
> \left[
> \mathscr{L}_S(\omega+u)
> \right],
> \qquad
> u \sim \mathrm{Unif}(B(0,\rho)).
> $$
>
> The quantity used in our generalization decomposition is bounded by local oscillation terms of the form
>
> $$
> \mathbb{E}
> \left[
> \left|\mathscr{L}_S(\omega+u)-\mathscr{L}_S^\rho(\omega)\right|
> \right],
> \qquad
> u \sim \mathrm{Unif}(B(0,\rho)),
> $$
>
> or, under Sobolev/Hölder regularity, by $C\rho^\alpha$. These formulas are already defined, but we will rewrite the derivation to remove the ambiguity noted by the reviewer.
>
> ### Notation corrections
>
> We thank the reviewer for pointing out the notation issues. On page 8, $\omega_f(\rho)$ denotes the modulus of continuity of a generic function $f$. We will make this definition explicit. We will also consistently use $\mathscr{L}_S$ and $\mathscr{L}_D$ for empirical and population losses, and correct inconsistent occurrences of $L_S$, $L_D$, or other variants.
>
> ### Experimental settings
>
> The models used in our experiments are specified in Table 1, including ResNet-20, ResNet-56, ResNext-29-32x4d, WRN-28-10, and PyramidNet-110. All compared methods, including SOTA baselines and LSAM, are trained with the same loss function. We keep the loss fixed across methods to ensure a fair comparison, so differences are attributable to the optimization method rather than changes in the training objective.
>
> ### Figures in Section 5
>
> The figures in Section 5 are arranged in pairs. Figure 2 reports performance over 200 epochs on CIFAR-10, while Figure 3 zooms in on epochs 150--200. Figure 4 and Figure 5 form a pair for CIFAR-100, and the same structure is used for Figures 6--7 and 8--9. The second figure in each pair makes late-stage differences more visible. We will revise captions and text to make this design explicit, and encourage readers to refer to the latter figure in each pair when comparing final-stage results.
>
> ### Role of $k$ in Algorithm 1 and Section 5.4
>
> The parameter $k$ is the number of Monte Carlo samples used to approximate the local average gradient in Algorithm 1. Larger $k$ reduces estimator variance, while $\rho$ controls the smoothing scale. We will state this before Section 5.4 and revise the discussion to separate their effects: $k$ mainly affects estimator stability, whereas $\rho$ controls smoothing and the bias--variance tradeoff.
>
> ### Section 5.6 and perturbation radius
>
> The subtitle of Section 5.6 is "Effect of Perturbation Radius and Optimization Hyperparameters". The first part focuses on perturbation radius $\rho$, while the second analyzes learning rate and weight decay. We will clarify this structure in the opening paragraph so that the transition is easier to follow.
>
> ### Figures and discussion in Section 5.4
>
> Section 5.4 already discusses how to interpret the corresponding figures. It explains the role of $\rho$ and $k$, and relates accuracy and loss curves to LSAM stability under local averaging.
>
> The figures in Section 5.4 are not intended to claim that the theoretical smoothing bias
>
> $$
> \left|\mathscr{L}_S^\rho(\omega)-\mathscr{L}_S(\omega)\right|
> $$
>
> is directly measured from the plots. Rather, they provide empirical evidence showing how choices of $\rho$ and $k$ affect training accuracy and loss. The discussion connects these trends with the theoretical role of local averaging.
>
> Some curves appear visually close because LSAM is stable across a range of hyperparameter choices, consistent with the robustness of the local-average objective. To avoid ambiguity, we will revise the text to point to relevant curves, epoch ranges, and metrics when comparing values of $\rho$ and $k$

---

> > ### Comment · Reviewer_AkMq · 2026-05-07
> >
> > Thanks for the clarifications. I still have some questions:
> >
> > 1. Clarification of the term $\overline{\mathscr{L}}_S^\rho(\omega)$: I understand the definition of it. What I do not understand is how you derived eqs (1). What I can obtain is simply $\mathbb{E}_u[\mathcal{L}_S(\omega + u) - \mathcal{L}_S(\omega)] \le \mathbb{E}_u [|\mathcal{L}_S(\omega + u) - \mathcal{L}_S(\omega)|]$".
> >
> > 2. Experimental settings: I would appreciate it if you could mention what loss function(s) you used in the experiments. This is important information for readers.
> >
> > 3. Figures in Section 5: I just want to emphasize that many the figures in Section 5.4 to Section 5.6 are crowded with lines of different colors and are very hard to read, including many of the "zoomed-in" figures such as Figure 7. So I will wait for the revised captions and text.
> >
> > 4. Role of $k$ in Algorithm 1 and Section 5.4: Thanks for the clarification. So please change the \textbf{capitalized} "K" in these sections to $k$.
> >
> > 5. Section 5.6 and perturbation radius: Could you please point out explicitly where you did discussed the effect of perturbation radius? As far as I can see, Figure 10 & 11 and related text mainly analyze learning rate for fixed perturbation radius, while Figure 12 & 13 and related text mainly analyze weight decay for fixed perturbation radius. I think the authors want to compare how the change of perturbation radius affects the result for different learning rates and/or weight decays. In this case it might be better to plot the loss & accuracy figures for different perturbation radius, with fixed learning rate/weight decay.

---

> > > ### Author Response · Authors · 2026-05-07
> > > **Response to Reviewer AkMq**
> > >
> > > ## Reply to 1
> > > Thank you for the clarification. You are right that the current wording around Eq. (21) is not precise if it is read as a direct consequence of Jensen’s inequality alone. Jensen’s inequality directly gives
> > >
> > > $$
> > > \left|
> > > \overline{\mathcal L}_S^\rho(\omega)-\mathcal L_S(\omega)
> > > \right|=
> > > \left|
> > > \mathbb E_u[\mathcal L_S(\omega+u)-\mathcal L_S(\omega)]
> > > \right|
> > > \le
> > > \mathbb E_u
> > > \left[
> > > |\mathcal L_S(\omega+u)-\mathcal L_S(\omega)|
> > > \right].
> > > $$
> > >
> > > Thus, Jensen directly controls the smoothing bias between the pointwise empirical loss and its local average through the center-based local oscillation.
> > >
> > > Our intended argument is based on the Sobolev/Hölder flatness framework introduced earlier in the paper. In the introduction, we state that the relevant points are characterized by local Sobolev regularity, which yields quantitative Hölder-flatness and explicit control of the smoothing bias. In Section 3, we also discuss that the local regularity of \(\mathcal L_S\) is translated into the generalization bound through the local averaging term.
> > >
> > > Under this local Hölder condition, for almost every regular point \(\omega\), there exist constants \(C_\omega>0\) and \(\alpha\in(0,1]\) such that
> > >
> > > $$
> > > |\mathcal L_S(\omega+u)-\mathcal L_S(\omega)|
> > > \le
> > > C_\omega\|u\|^\alpha .
> > > $$
> > >
> > > Since \(u\sim{\rm Unif}(B_\rho)\) satisfies \(\|u\|\le\rho\), the smoothing bias is bounded by
> > >
> > > $$
> > > \left|
> > > \overline{\mathcal L}_S^\rho(\omega)-\mathcal L_S(\omega)
> > > \right|
> > > \le
> > > C_\omega\rho^\alpha .
> > > $$
> > >
> > > When the Hölder regularity is induced by Sobolev embedding, one may take \(\alpha=1-d/p\) under the condition \(\mathcal L_S\in W^{1,p}_{\rm loc}(\mathbb R^d)\), \(p>d\).
> > >
> > > Importantly, the local-average oscillation used in our flatness interpretation is also controlled under the same Sobolev/Hölder condition. Indeed, by the triangle inequality,
> > >
> > > $$
> > > |\mathcal L_S(\omega+u)-\overline{\mathcal L}_S^\rho(\omega)|
> > > \le
> > > |\mathcal L_S(\omega+u)-\mathcal L_S(\omega)|
> > > +
> > > |\mathcal L_S(\omega)-\overline{\mathcal L}_S^\rho(\omega)|.
> > > $$
> > >
> > > Taking expectation over \(u\) and using the two bounds above gives
> > >
> > > $$
> > > \mathbb E_u
> > > \left[
> > > |\mathcal L_S(\omega+u)-\overline{\mathcal L}_S^\rho(\omega)|
> > > \right]
> > > \le
> > > 2C_\omega\rho^\alpha .
> > > $$
> > >
> > > In the Sobolev-induced case, this becomes
> > >
> > > $$
> > > \mathbb E_u
> > > \left[
> > > |\mathcal L_S(\omega+u)-\overline{\mathcal L}_S^\rho(\omega)|
> > > \right]
> > > \le
> > > 2C_\omega\rho^{1-d/p}.
> > > $$
> > >
> > > Therefore, the intended message remains unchanged: local Sobolev/Hölder regularity controls both the smoothing bias and the mean oscillation around the local average, which is the average-case flatness quantity associated with LSAM. We will revise Eq. (21) and the surrounding text to make this logical order explicit: Jensen first controls the smoothing bias through center-based oscillation, while the mean oscillation around the local average is controlled under the same Sobolev/Hölder regularity framework. This avoids suggesting that the latter follows directly from Jensen alone.
> > > ## Reply to 2
> > > In accordance with the principle of consistency, we use cross-entropy loss.
> > > ## Reply to 3
> > > Thank you for pointing this out. The colored bars in the figures represent error bars. In each figure, we present and compare five different scenarios, and the error bars are obtained from experiments with different random seeds.
> > > ## Reply to 4
> > > We will standardize the characters.
> > > ## Reply to 5
> > > We apologize for the lack of clarity. In Section 5.6, we conducted a detailed comparison using rho values of 0.05 and 0.02. Other rho values are shown in Figures 6 through 9. The conclusions reached in this section are based on the preceding discussion and the detailed analysis in Section 5.6. To avoid confusion, we will clarify this in th

---

### Decision · Action_Editor_oZ1z · 2026-06-15

**Recommendation:** Accept with minor revision

**Additional Comments:**

Please take into account all the typos and corrections identified by reviewers (particularly AkMq). Please strongly consider restructuring the text of the paper to move proofs/additional experiments, figures to an appendix. The current paper is very long at 26 pages and the message will land quicker if the authors separate their results and proofs. This is a minor revision since reviewers have not requested changes to the core arguments/theorems.

**Audience:**

Yes

**Audience Explanation:**

This paper proposes new Sharpness Aware Minimization training algorithms that can in principle be combined with other optimizers.

**Claims And Evidence:**

Yes

**Claims Explanation:**

All reviewers agree that the paper makes a useful modification to Sharpness Aware Minimization through the introduction of local average loss instead of the training loss. The proposed method is empirically shown to perform better on small models, and the authors conjecture that the landscape of larger models may already be smooth.